# A Hierarchical Geometric Observation Interface for Spatial Planning in Reinforcement Learning

## Abstract

In reinforcement learning (RL), spatial planning is often mediated through rasterized observations processed by convolutional networks, even when the underlying task is continuous and geometric. This discretization can introduce aliasing and obscure topological structure, increasing the difficulty of the spatial problem. We study a hierarchical set-valued geometry-first observation interface for sparse-reward navigation that operates directly on triangulated obstacle geometry. This interface uses learned multi-token aggregation to compress variable-sized geometry into a bounded fixed-size representation while preserving local spatial structure relevant for spatial decision making. In a controlled goal-conditioned point-navigation setting with a fixed RL backbone, we compare it against raster–CNN baselines across bounded and unbounded procedural training regimes. The empirical results of our work demonstrate that its advantage is most pronounced under continual exposure to newly generated environments, where the agent must learn reusable spatial structure rather than rely on memorizing a fixed environment support.

## 1 Introduction

In reinforcement learning (RL) for spatial planning, the observation interface determines which geometric regularities a policy can readily exploit. This is especially important in continuous domains, where successful behavior depends on spatial relationships that are metric, relational, and not naturally discrete.

Despite this, RL still commonly approaches such problems by processing rasterized observations with convolutional networks (Espeholt et al., 2018; Sharma et al., 2025). This choice is practical, but it creates a representational mismatch: continuous spatial structure is presented through a discretized, image-like proxy. Rasterization can introduce aliasing, impose resolution-dependent trade-offs, and tie spatial reasoning to perceptual encoding. In sparse-reward navigation, where exploration and long-horizon credit assignment are already challenging, this coupling can make it difficult to tell whether failure arises from planning itself or from the representation used to support it (Pignatelli et al., 2023; Cetin et al., 2022).

A natural alternative is to expose geometry directly—increasingly plausible given modern sensing and mapping pipelines—and to study control on a representation that preserves the structure most relevant for spatial decision-making. In this work, we adopt a geometry-first view and instantiate it using triangles as obstacle primitives. Triangles provide a lightweight, continuous representation with a fixed internal structure, while preserving metric information and accommodating complex topology. Direct geometry alone, however, does not solve the representation problem: a triangulated scene is a variable-size unordered set of primitives, so the encoder must handle permutation invariance, changing cardinality, and potentially large primitive counts (Zaheer et al., 2017; Lee et al., 2019b). This rules out naive sequence encodings and makes simple global pooling only a partial solution, since effective control may depend on interactions among nearby primitives before compression. At the other extreme, unrestricted global self-attention over all primitives quickly becomes computationally impractical at realistic scene densities. Graph-based processing offers a related bias, but long-range structure must then emerge through repeated message passing rather than explicit local aggregation. The central challenge is therefore not only how to expose geometry to the policy, but also how to do so in a way that remains tractable and geometrically well-aligned. To address this, we instantiate a

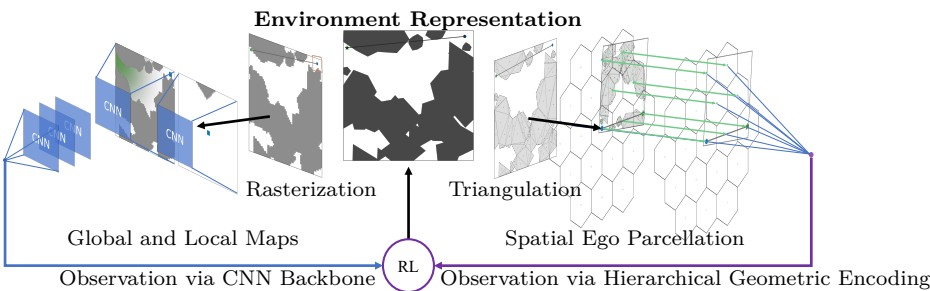

**Environment Representation**

Figure 1: **Two Observation Interfaces for the same navigation environment.**

hierarchical geometry-first observation interface for sparse-reward navigation. The design organizes triangulated obstacle geometry into a hex-local set-of-sets representation and aggregates it hierarchically into a fixed-dimensional state. This turns geometric observation into a learned compression problem: multiple aggregation seeds form a bounded multi-faceted summary of local and global geometry, while preserving local metric structure and avoiding the cost of flat global self-attention. Although its ingredients are individually familiar, their synthesis yields a geometry-first interface that is both tractable for RL and well suited to controlled evaluation.

The goal of this paper is therefore not only to introduce a tractable geometry-first observation interface, but also to test when it becomes useful. For spatial RL to scale beyond fixed map collections, agents should benefit from increasing environmental diversity rather than merely fit a finite set of training layouts—which makes broad procedural coverage, with sparse rewards and geometry drawn from multiple environment families, the natural stress test.

We study this question in a controlled goal-conditioned point-navigation setting under a fixed RL backbone. The training setup spans multiple complex procedural distributions, and evaluation includes both in-distribution and held-out generator families. Using raster–CNN baselines as the main comparison, we isolate the effect of the observation interface from optimization and actor–critic confounds. Importantly, the raster baselines are not restricted to binary occupancy but are enriched with continuous distance cues, so the comparison asks whether explicit geometric structure adds value beyond metric information alone. We further distinguish finite-support training from sustained procedural generation. The results show a clear pattern: the geometry-first interface is strongest when training continually exposes the agent to new layouts, while the gap narrows when learning is restricted to a fixed finite map pool.

Our contributions are as follows:

- We instantiate a hex-local hierarchical geometry-first interface that uses learned multi-token aggregation to compress triangulated obstacle geometry into a bounded fixed-dimensional set-of-sets observation.

- We show that hierarchical aggregation makes geometry-first RL computationally tractable at realistic primitive counts, where flatter geometric attention models become impractical.

- Through a controlled comparison with raster–CNN baselines under a fixed RL backbone, we show that the benefit of the proposed interface is most pronounced under sustained procedural novelty and smaller under bounded training.

## 2 Related Work

**Abstraction and generalization in spatial RL.** In embodied navigation and related spatial RL settings, a parallel line of work has introduced richer abstractions beyond raw raster inputs, including semantic maps, topological memories, and scene-graph-like representations (Chaplot et al., 2020; Singh et al., 2023). These can improve long-horizon reasoning and semantic generalization, but they typically shift the inductive bias

away from direct metric geometry and often rely on additional semantic structure or auxiliary signals. By contrast, vectorized spatial representations are common in adjacent areas such as motion forecasting, yet remain relatively uncommon as the primary interface for RL policies (Gao et al., 2020). Our focus is different: we study whether operating directly on structured geometry helps RL learn spatial reasoning under distribution shift.

**Image encoders in visual RL.** In modern RL, encoders are often deep residual CNNs, with IMPALA-style backbones forming a common design choice (Espeholt et al., 2018). This approach is practical and often effective, but it ties control to an image-processing pipeline shaped by discretization, resolution, and the compression of spatial feature maps before the policy head. Recent work has argued that this encoder-to-head interface can itself become a scaling bottleneck in pixel-based RL, and that replacing flattening with global average pooling can improve performance and generalization (Trumpp et al., 2025; Sokar & Castro, 2025). We share the focus on the encoder-to-policy interface, but study it in a deliberately controlled setting: a fixed, single-stream actor–critic backbone, without recurrence or distributed training. Our work departs from the raster–CNN paradigm by operating directly on structured geometry: the input tokens are geometric primitives organized through an explicit hex-local hierarchy, yielding a more interpretable and domain-specific bottleneck than generic image patches.

**Set and geometric encoders.** Since our observation is a variable-size unordered collection of geometric primitives, permutation-invariant set processing is a natural starting point. Deep Sets formalized invariant learning on sets; PointNet and PointNet++ (Qi et al., 2017a;b) showed that shared encoders with symmetric aggregation and local hierarchy can be effective on unordered geometric inputs; and Set Transformer extended this family with attention-based interaction modeling (Zaheer et al., 2017; Lee et al., 2019b). Graph neural networks provide a related relational bias, but long-range context typically emerges only through repeated message passing, which introduces additional design choices and potential propagation bottlenecks (Rusch et al., 2023; Black et al., 2023). Our method builds on this broader literature, but differs from flat set encoding and generic graph propagation by imposing locality explicitly through a hexagonal spatial hierarchy and bounded hierarchical aggregation.

## 3 Method

We formulate the point-to-point navigation problem as a goal-conditioned RL task. We then construct the proposed hierarchical observation interface from the geometric environment representation and specify the tensor form of the resulting observation space. Finally, we describe how this structured observation is ingested by the hierarchical geometric encoder.

### 3.1 Problem Formulation

Spatial planning may be viewed as a geometric problem in which an agent traverses a spatial domain $D$. Let $D \coloneqq [0,1]^2 \subset \mathbb{R}^2$ be bounded and planar. We define a *map* $G \in \mathfrak{G}$ as a finite collection of spatial elements in $D$. For a given map $G$, we write $S(G) \subset D$ for the union of all solid obstacle polygons, and define the corresponding free space as $F(G) \coloneqq D \setminus S(G)$. Since $S(G)$ is polygonal, it admits a finite triangulation $\mathcal{T}(G) = \{T_r\}_{r=1}^{N_T}$, whose triangles cover the obstacle region $S(G)$ up to numerical tolerance.

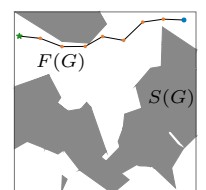

Figure 2: Piecewise-linear trajectory in a polygonal environment.

Within this geometric abstraction, we formulate the navigation task as a goal-conditioned Markov decision process (gc-MDP) $\mathcal{M} = \langle \hat{\mathcal{S}}, \mathcal{A}, \mathcal{P}, r, \gamma, \rho_0 \rangle$, where $\hat{\mathcal{S}}$ is the augmented state space, $\mathcal{A}$ the action space, $\mathcal{P}$ the transition kernel, $r$ the reward function, $\gamma \in (0,1)$ the discount factor, and $\rho_0$ the reset distribution. In each episode, both the map instance $G \in \mathfrak{G}$ and the goal location $\mathbf{g} \in F(G)$ remain fixed, while only the agent position evolves over decision steps. The augmented state is therefore $\hat{s}_t \coloneqq (\mathbf{x}_t, \mathbf{g}, G) \in \hat{\mathcal{S}}$, where $\hat{\mathcal{S}} \coloneqq \bigcup_{G \in \mathfrak{G}} \big(F(G) \times F(G) \times \{G\}\big)$ and $\mathbf{x}_t \in F(G)$ denotes the agent position at step $t$. The corresponding geometric trajectory is the piecewise-linear path ob-

tained by interpolating between successive sampled positions. The action space is $\mathcal{A} = [-1, 1]^2 \subset \mathbb{R}^2$, whose elements are interpreted as line segments in $G$. For a fixed maximum step length $\alpha > 0$, each action $\mathbf{a}_t \in \mathcal{A}$ proposes the next position $\tilde{\mathbf{x}}_{t+1} = \mathbf{x}_t + \alpha \mathbf{a}_t$. The move is accepted if the segment $[\mathbf{x}_t, \tilde{\mathbf{x}}_{t+1}]$ lies entirely in free space, in which case $\mathbf{x}_{t+1} = \tilde{\mathbf{x}}_{t+1}$; otherwise the agent remains at $\mathbf{x}_t$ and the episode terminates. The agent acts on observations $\mathbf{o}_t = \phi(\hat{s}_t)$ rather than on the full augmented state directly, where $\phi$ denotes the observation function. We consider a sparse-reward setting in which reward is given only on reaching the goal, namely $r(\hat{s}_t, \mathbf{a}_t, \hat{s}_{t+1}) = \mathbf{1}[\|\mathbf{x}_{t+1} - \mathbf{g}\|_2 < \epsilon]$ for a fixed threshold $\epsilon > 0$. The objective is to learn a policy $\pi(\mathbf{a}_t \mid \mathbf{o}_t)$ that maximizes the expected discounted return $J(\pi) = \mathbb{E}_{\tau \sim (\rho_0, \mathcal{P}, \phi, \pi)}[\sum_{t=0}^{\infty} \gamma^t r(\hat{s}_t, \mathbf{a}_t, \hat{s}_{t+1})]$.

## 3.2 The Observation Interface

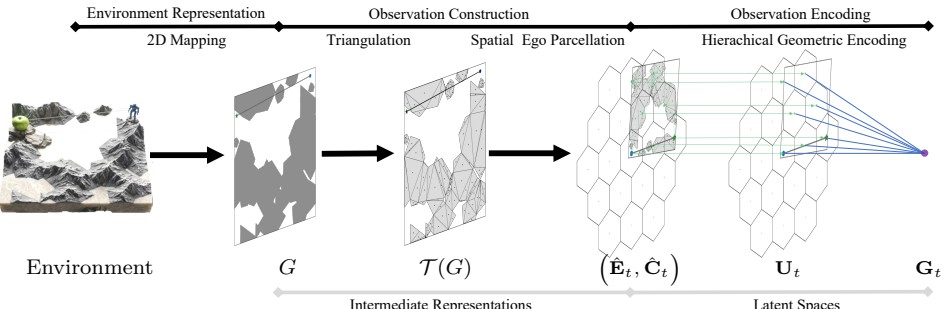

Figure 3: Full Observation Interface

### 3.2.1 Observation Construction

We construct an observation space that presents the surrounding geometry as a hierarchy of local geometric subsets.

**Spatial Ego Parcellation** The observation is constructed in an ego frame aligned with the agent–goal axis: the $x$-axis points toward the goal, and all geometry is expressed relative to this frame. Let $\theta_t$ denote this heading, and $\mathbf{R}_{\theta_t}$ the ego-to-world rotation matrix. A hexagonal lattice, fixed in this ego frame, partitions the surrounding plane into spatial cells based on the $A_2$ (triangular) Bravais lattice, yielding regular hexagonal cells (Conway & Sloane, 1999) indexed by axial coordinates $(q, r) \in \mathbb{Z}^2$. The lattice scale $s > 0$ — defined as the hexagon circumradius — sets the spatial resolution and serves as the normalization scale throughout the representation. Cell centers follow the standard pointy-top mapping:

$$\mathbf{c}_{q,r} = \begin{bmatrix} \sqrt{3}\, s\, (q + \frac{1}{2}r) \\ \frac{3}{2}\, s\, r \end{bmatrix}. \tag{1}$$

The receptive field is bounded by restricting to an $R$-ring neighborhood using the discrete hex distance $d_{\text{hex}}(q, r) = \frac{|q| + |r| + |q+r|}{2}$, giving at most $N_{\text{hex}} = 1 + 3R(R+1)$ cells. Hexagons provide an equal-area, nearly isotropic partition of the plane (Middleton & Sivaswamy, 2005). The set of active cell centers $C_R := \left\{ \mathbf{c}_{q,r} : (q, r) \in \mathbb{Z}^2, d_{\text{hex}}(q, r) \leq R \right\}$ is represented as the dense tensor

$$\mathbf{C}_t := \left[ \mathbf{c}_{q_1, r_1}^\top, \ldots, \mathbf{c}_{q_{N_{\text{hex}}}, r_{N_{\text{hex}}}}^\top \right]^\top \in \mathbb{R}^{N_{\text{hex}} \times 2} \tag{2}$$

with binary mask $\boldsymbol{m}_t^{\text{cell}} \in \{0, 1\}^{N_{\text{hex}}}$ indicating occupied cells. Each primitive is assigned to a cell by its ego-frame centroid via cube rounding, then re-centered and normalized by $s$ to yield dimensionless cell-local coordinates. The ego-frame state is summarized by the navigation vector $\mathbf{z}_t = \left[ \cos(\theta_t), \sin(\theta_t), x_t, y_t, \tilde{g}_{x,t}, \tilde{g}_{y,t}, \hat{L}_t \right]^\top \in \mathbb{R}^7$, where $(x_t, y_t)$ is the agent position in world coordinates, the ego-frame goal coordinates $(\tilde{g}_{x,t}, \tilde{g}_{y,t})[1]$ and the agent–goal Euclidean distance $\hat{L}_t = \|\mathbf{g} - \mathbf{x}_t\|_2 / s$ are normalized by the lattice circumradius $s$. The full egocentric construction is illustrated in App. A.1 (Fig. 12).

---

[1]Retained for generality, e.g., under agent-pose rather than goal-aligned frames.

**Geometric Primitives** We use triangles as the basic geometric primitive. Their fixed internal structure simplifies encoding while preserving continuous geometry. For each triangle $\hat{T} = \{a, b, c\}$ expressed in cell-local coordinates, we compute the triangle feature vector $\psi(\hat{T}) = \left[ a^\top, \ b^\top, \ c^\top, \ \bar{p}^\top, \ \|b-a\|_2, \ \|c-b\|_2, \ \|a-c\|_2, \ A(\hat{T}) \right] \in \mathbb{R}^{12}$, where $\bar{p} = (a + b + c)/3$ is the triangle centroid and $A(\hat{T})$ is the signed area. We store the resulting fixed-sized tensor of triangle feature vectors $\mathbf{E}_t$ together with a binary validity mask $\boldsymbol{m}_t^{\text{tri}} \in \{0, 1\}^{K_{\text{tri}}}$, which indicates which of the $K_{\text{tri}}$ entries are valid.

$$\mathbf{E}_t \in \mathbb{R}^{K_{\text{tri}} \times 12}. \tag{3}$$

To guarantee a fixed-size observation tensor, we retain at most $M$ active cells and at most $K_{\text{cell}}$ triangle feature vectors per cell[2]. This truncation yields $\hat{\mathbf{E}}_t \in \mathbb{R}^{M \times K_{\text{cell}} \times 12}$, $\hat{\boldsymbol{m}}_t^{\text{tri}} \in \{0, 1\}^{M \times K_{\text{cell}}}$, $\hat{\mathbf{C}}_t \in \mathbb{R}^{M \times 2}$, $\hat{\boldsymbol{m}}_t^{\text{cell}} \in \{0, 1\}^M$, together with $\mathbf{z}_t \in \mathbb{R}^7$. The resulting observation is

$$\mathbf{o}_t = \left( \hat{\mathbf{E}}_t, \ \hat{\boldsymbol{m}}_t^{\text{tri}}, \ \hat{\mathbf{C}}_t, \ \hat{\boldsymbol{m}}_t^{\text{cell}}, \ \mathbf{z}_t \right). \tag{4}$$

### 3.2.2 Observation Encoding

We encode the bounded hex-local set-of-sets representation with a two-stage hierarchical set encoder that first aggregates triangles within each retained hex cell and then aggregates the resulting cell-level summaries across cells into a fixed-dimensional latent representation. We use masked Pooling by Multihead Attention (PMA) (Lee et al., 2019a) as the aggregation operator at both levels, enabling multi-faceted learned summaries through multiple seed tokens that jointly form a compressed representation. PMA introduces $k$ learned seed vectors $S \in \mathbb{R}^{k \times d}$ and computes:

$$\text{PMA}_k(Z) = \text{MHA}(Q{=}S, \ K{=}Z, \ V{=}Z) \in \mathbb{R}^{k \times d}. \tag{5}$$

This layer maps unordered sets to invariant multi-token readouts without quadratic scaling. In our implementation, we employ a masked version of PMA. Although the tensors are padded to a fixed size, the number of valid entries varies per state; masked permutation-invariant pooling keeps the encoder independent of slot order and padding, and lets the capacity change at deployment time (Tab. 3, App. D.3). We describe the full pipeline using its conceptual steps:

**Primitive encoding.** A shared residual encoder is applied independently to each triangle descriptor: $\mathbf{V}_t = \text{Enc}_{\text{tri}}(\hat{\mathbf{E}}_t) \in \mathbb{R}^{M \times K_{\text{cell}} \times d_{\text{tri}}}$.

**Intra-cell aggregation.** For each retained cell, PMA with $S_{\text{tri}}$ learned seeds aggregates the unordered set of triangle features into a fixed number of cell-level tokens: $\mathbf{U}_t = \text{PMA}_{S_{\text{tri}}}(\mathbf{V}_t, \hat{\boldsymbol{m}}_t^{\text{tri}}) \in \mathbb{R}^{M \times S_{\text{tri}} \times d_{\text{cell}}}$.

**Global aggregation.** We inject spatial context by adding an encoded representation of each retained hex-cell center to its corresponding cell-level tokens. This requires the seed tokens produced by triangle aggregation to match the $d_{\text{cell}}$-dimensional center-feature space; in general, this can be achieved with a projection layer, while in our implementation, the projection is the identity map. $\widetilde{\mathbf{U}}_t = \mathbf{U}_t + \text{broadcast}_{S_{\text{tri}}}(\text{Enc}_c(\hat{\mathbf{C}}_t)) \in \mathbb{R}^{M \times S_{\text{tri}} \times d_{\text{cell}}}$. These tokens are then flattened across cells and seed slots and aggregated with a second PMA layer using the correspondingly broadcast cell-validity mask: $\mathbf{G}_t = \text{PMA}_{S_{\text{grid}}}\left( \text{flatten}(\widetilde{\mathbf{U}}_t), \ \text{broadcast}_{S_{\text{tri}}}(\hat{\boldsymbol{m}}_t^{\text{cell}}) \right) \in \mathbb{R}^{S_{\text{grid}} \times d_{\text{cell}}}$.

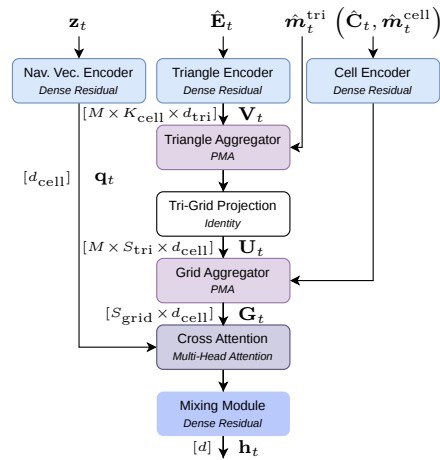

Figure 4: **Architecture of hierarchical geometric encoder (Hex-PMA).**

**Vector fusion.** The encoded navigation vector is used as a query in a cross-attention module over the global summary tokens, followed by a residual MLP that produces the latent state: $\mathbf{q}_t = \text{Enc}_z(\mathbf{z}_t)$, $\mathbf{h}_t = \text{MLP}_{\text{mix}}(\text{CrossAttn}(\mathbf{q}_t, \mathbf{G}_t))$.

---

[2]Although this step could easily host an informed heuristic, we deliberately use a task-agnostic, deterministic truncation.

# 4 Experimental Setup

Our experiments are designed to vary training and evaluation conditions in a controlled way while isolating the effect of the observation interface. We first introduce the *Environment Generation Controls* (Sec. 4.1), then the *Training and Evaluation Protocols* (Sec. 4.2), and finally the *Comparison Setup* (Sec. 4.3) that holds the RL backbone fixed across methods. This organization makes explicit how the task distribution is controlled, how generalization is evaluated, and how the representation question is isolated.

## 4.1 Environment Generation Controls

We generate planar environments from procedural map generators,[3] each defining a qualitative family. Representative samples from all generators are shown in Fig. 5. We control variation along three axes: generator family, geometric complexity, and task difficulty. Here, geometric complexity is controlled through a primitive budget[4]; the precise generator-side parameterization is given in App. C. Task difficulty is controlled through the start–goal sampling distribution. A summary of all generators, including geometric complexity and per-cell primitive density statistics[5], is provided in Tab. 3 (App.). The training families are deliberately chosen so that observation truncation actually occurs also during training.

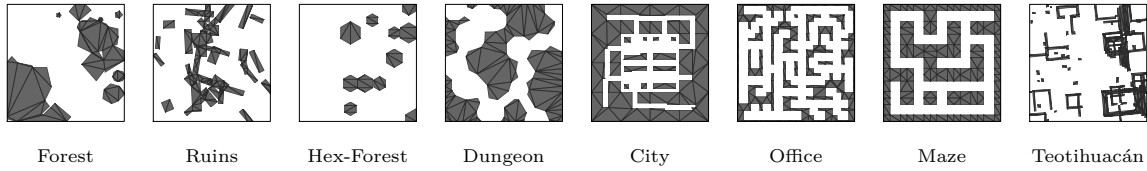

| Forest | Ruins | Hex-Forest | Dungeon | City | Office | Maze | Teotihuacán |

Figure 5: Samples from all eight procedural map generators used in this work.

## 4.2 Training and Evaluation Protocols

We study two training regimes. In the *unbounded* regime, training maps are drawn from a continually expanding procedural stream. In the *bounded* regime, training is restricted to a fixed finite set of maps. Evaluation varies along two axes: *difficulty*, controlled by the reset-difficulty parameter $t_{\text{diff}}$ (defined in App. C, Fig. 18), and *provenance* of the test map. We evaluate on three map distributions: ID–Seen, generated by the training generators with seeds encountered during training; ID–Unseen, generated by the same generators with held-out seeds; and OOD, generated by held-out generators. Taken together, the protocol may be viewed as a matrix indexed by training regime, evaluation distribution, and fixed evaluation difficulty.

## 4.3 Comparison Setup

**Shared Training and Evaluation Setup.** All agents are trained with SAC (Haarnoja et al., 2018) under identical replay, training budget, discount, and curriculum settings, and share the same modular actor–critic template with fixed policy and twin critic heads. Methods differ only in their observation interface—observation function plus encoder—which jointly map the environment state into a common latent space. This isolates the effect of the observation interface from optimization and downstream architecture confounds. We evaluate three aspects: training dynamics, final navigation performance, and computation. Training progress is measured by curriculum progression rather than raw training success, since task difficulty changes over the course of learning. Final performance is reported using success rate and path efficiency over successful episodes relative to an oracle planner[6]. Computational efficiency is assessed through inference latency, update cost, and parameter count. Full hyperparameter settings (Tabs. 4 and 5), architectural details, and metric definitions are provided in the appendix.

---

[3]Deterministic mappings from random seeds to environment geometries.

[4]the maximum number of triangular primitives used to represent an environment

[5]Measured under the observation-construction hyperparameters used in the main experiment.

[6]Shortest paths are computed with continuous A*, with a discrete grid-based A* fallback when numerical instabilities arise.

**Observation Functions and Encoders.** The raster baselines, IMPALA-CNN (Espeholt et al., 2018) (hereafter IMPALA) and IMPOOLA (Trumpp et al., 2025), receive a standard two-stream raster interface consisting of a global map and a local crop, both processed by convolutional encoders; Importantly, these inputs are not limited to binary occupancy, but also include signed-distance-field, agent-position, and goal-position channels, so the comparison tests direct geometric structure against raster observations already enriched with continuous metric cues. The raster baselines also receive the same goal-relative orientation as the geometric interface: the local crop is rotated so that the agent–goal direction aligns with the positive $x$-axis, and all methods receive the same navigation-vector format (App. B.2). Orientation handling is therefore matched across interfaces and does not confound the comparison. To isolate encoder effects under matched geometric inputs, we also consider HEX-ATTN, which uses the same hexagonal spatial parcellation and tokenization as HEX-PMA but replaces PMA with self-attention and CLS-based global aggregation (Vaswani et al., 2017; Devlin et al., 2019). Finally, the global TRANSFORMER baseline uses the same triangulated geometry without hex spatial parcellation and processes it with a standard self-attention encoder and the same CLS-based global aggregation (Vaswani et al., 2017; Devlin et al., 2019).[7]

## 5 Results and Analysis

The main comparison evaluates HEX-PMA against raster–CNN baselines on the full task setting, whereas the geometric encoder ablations keep the same comparison principle but move to a simplified procedural setting with reduced generator and primitive budgets in order to make alternative geometric encoders computationally feasible. Full architectural and training details are provided in App. B and App. C. We organize the analysis around two questions: First, does the advantage of the geometric interface depend on the training regime?—the focus of the main comparison; Second, does bounded hierarchical aggregation retain the benefits of geometric processing without incurring the cost of full global attention?—the focus of the geometric encoder ablations.

### 5.1 Main comparison: Hex-PMA versus raster baselines

The main comparison is conducted on the complex procedural generators *Ruins* and *Forest*, which stress the observation interface through clutter, narrow passages, and multiscale obstacle structure. Within this setting, we compare HEX-PMA against two raster–CNN baselines, IMPALA and IMPOOLA.[8] Training dynamics are summarized through aggregate metrics in the main text, with full learning curves deferred to App. D.2.

#### 5.1.1 Unbounded training

| Method | Success Rate | | Curriculum Mean | |
|---|---|---|---|---|
| | AUC↑ | Asympt.↑ | AUC↑ | Asympt.↑ |
| HEX-PMA | **74.3 ± 0.7** | **77.1 ± 0.7** | **0.898 ± 0.026** | **0.999 ± 0.000** |
| IMPOOLA | 69.7 ± 0.1 | 70.0 ± 0.1 | 0.443 ± 0.052 | 0.475 ± 0.077 |
| IMPALA | 69.6 ± 0.1 | 69.9 ± 0.1 | 0.220 ± 0.021 | 0.226 ± 0.053 |

Table 1: Unbounded training dynamics (higher is better). Full curves in App. D.2.

In the unbounded regime, training maps are drawn from a continually expanding procedural stream rather than a fixed finite pool. This is a demanding test of representation quality, since the agent must repeatedly adapt to new layouts instead of improving in a stable environment. Tab. 1 summarizes training dynamics in the unbounded regime. The clearest separation appears in the curriculum metrics: HEX-PMA attains by far the highest curriculum AUC and reaches an asymptotic curriculum mean difficulty of nearly 1.0, indicating that it successfully progresses to the highest difficulty levels. By contrast, the raster baselines

---

[7]In this sense, HEX-ATTN is the TRANSFORMER baseline augmented with hexagonal spatial parcellation.

[8]HEX-PMA and IMPOOLA are closely matched in parameter count. IMPALA uses the same overall convolutional structure as IMPOOLA, but its terminal flattening stage yields a larger parameter count than global pooling; see Sec. 5.1.3.

remain confined to much lower curriculum levels throughout training. This gap is also reflected in success-rate AUC and asymptotic success, where HEX-PMA again performs best despite dealing with substantially harder tasks.

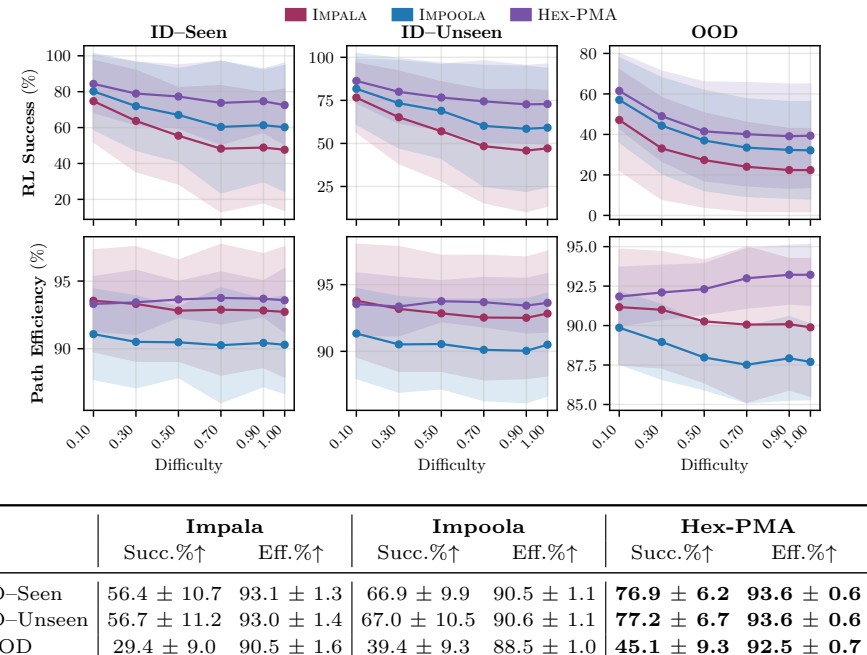

|  | Impala | | Impoola | | Hex-PMA | |
|---|---|---|---|---|---|---|
|  | Succ.%↑ | Eff.%↑ | Succ.%↑ | Eff.%↑ | Succ.%↑ | Eff.%↑ |
| ID–Seen | 56.4 ± 10.7 | 93.1 ± 1.3 | 66.9 ± 9.9 | 90.5 ± 1.1 | **76.9 ± 6.2** | **93.6 ± 0.6** |
| ID–Unseen | 56.7 ± 11.2 | 93.0 ± 1.4 | 67.0 ± 10.5 | 90.6 ± 1.1 | **77.2 ± 6.7** | **93.6 ± 0.6** |
| OOD | 29.4 ± 9.0 | 90.5 ± 1.6 | 39.4 ± 9.3 | 88.5 ± 1.0 | **45.1 ± 9.3** | **92.5 ± 0.7** |

Figure 6: **Unbounded training.** Success rate and path efficiency across evaluation splits and difficulty levels; table reports averages over difficulty.

This training time performance difference carries over to the evaluation. Across ID–Seen, ID–Unseen, and OOD evaluation, all methods degrade as difficulty increases, but HEX-PMA consistently attains the strongest success rates (Fig. 6). The gap is modest at low difficulty and widens substantially at intermediate and high difficulty, indicating that the benefit of the geometric interface becomes most visible when navigation requires longer-horizon, obstacle-aware planning. This pattern is especially clear under OOD evaluation, where HEX-PMA degrades more gradually than either raster baseline. The averages in Fig. 6 show the same ordering across all three evaluation distributions. Path efficiency shows a complementary pattern. Whereas the raster baselines decline mildly with increasing difficulty, HEX-PMA remains nearly flat and, on OOD maps, even improves slightly with difficulty. At the easiest settings, IMPALA can match or slightly exceed HEX-PMA on in-distribution path efficiency, but this advantage disappears once planning becomes more global. In addition, HEX-PMA exhibits visibly lower spread across random seeds, suggesting more stable optimization under sustained procedural novelty.

### 5.1.2 Bounded training

|  | Success Rate | | Curriculum Mean | |
|---|---|---|---|---|
| **Method** | AUC↑ | Asympt.↑ | AUC↑ | Asympt.↑ |
| HEX-PMA | **87.3 ± 3.0** | **92.1 ± 2.3** | **0.937 ± 0.027** | **1.000 ± 0.000** |
| IMPOOLA | 83.6 ± 2.1 | 89.0 ± 1.1 | 0.893 ± 0.033 | **1.000 ± 0.000** |
| IMPALA | 82.4 ± 1.6 | 86.7 ± 1.1 | 0.918 ± 0.022 | 0.985 ± 0.022 |

Table 2: Bounded training dynamics (higher is better). Full curves in App. D.2.

In the bounded regime, training is restricted to a fixed finite set of maps generated from the same *Ruins* and *Forest* families[9]. This setting is less demanding from a generalization perspective, since the agent can repeatedly revisit the same environment support throughout training. It adopts the finite-support setting used in many procedural RL generalization benchmarks, where performance on held-out environments from the same support is often treated as generalization. We include this regime because it is also practically relevant when only a finite sample of environments is available during training.

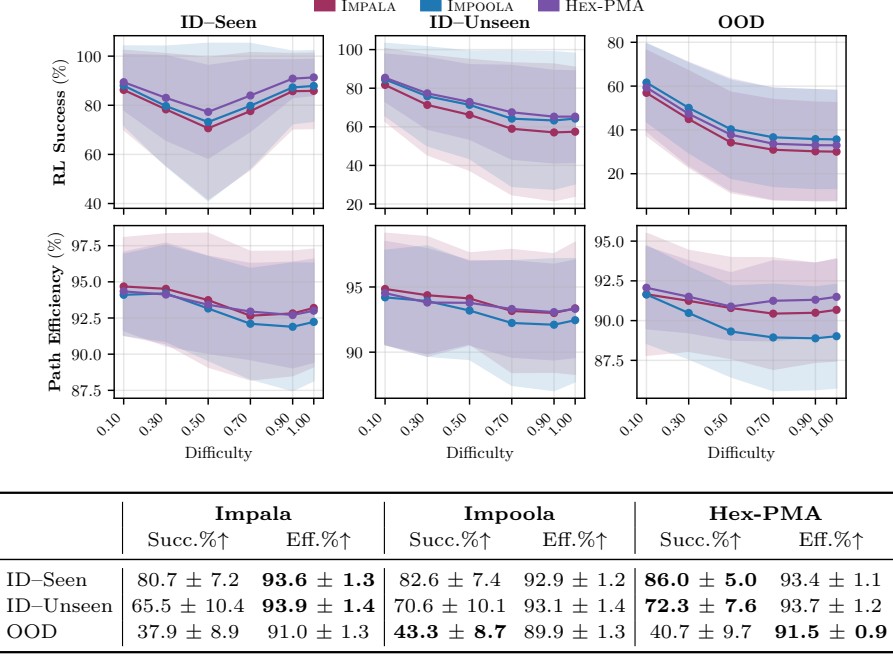

| | Impala | | Impoola | | Hex-PMA | |
| | Succ.%↑ | Eff.%↑ | Succ.%↑ | Eff.%↑ | Succ.%↑ | Eff.%↑ |
|---|---|---|---|---|---|---|
| ID–Seen | 80.7 ± 7.2 | **93.6 ± 1.3** | 82.6 ± 7.4 | 92.9 ± 1.2 | **86.0 ± 5.0** | 93.4 ± 1.1 |
| ID–Unseen | 65.5 ± 10.4 | **93.9 ± 1.4** | 70.6 ± 10.1 | 93.1 ± 1.4 | **72.3 ± 7.6** | 93.7 ± 1.2 |
| OOD | 37.9 ± 8.9 | 91.0 ± 1.3 | **43.3 ± 8.7** | 89.9 ± 1.3 | 40.7 ± 9.7 | **91.5 ± 0.9** |

Figure 7: **Bounded training.** Success rate and path efficiency across evaluation splits and difficulty levels; table reports averages over difficulty.

Tab. 2 shows that the training gap between methods is substantially smaller than in the unbounded regime. All three reach high curriculum levels by the end of training, and although HEX-PMA still leads on the aggregate training metrics, its advantage over the raster baselines is no longer decisive. The same compression of differences appears at evaluation time (Fig. 7). On the in-distribution splits, all methods remain much closer than in the unbounded case. HEX-PMA still retains a modest lead in success on ID–Seen and ID–Unseen evaluation, but the separation is smaller than under continual procedural novelty. Differences in path efficiency are also small throughout the in-distribution evaluations. OOD performance is more mixed: IMPOOLA attains slightly higher average OOD success, while HEX-PMA remains the most balanced across the three evaluation distributions; path-efficiency differences between methods are small in this regime and fall within seed variability (Fig. 7). This contrast with the unbounded regime is important. One interpretation is that, once the training distribution is bounded, the raster baselines can exploit recurring regularities in the finite map pool more effectively, which reduces the visible advantage of the stronger geometric inductive bias. In this sense, bounded training does not remove the representational difference between the interfaces, but it partially masks it. This may help explain why Procgen-like finite-support benchmarks can make raster CNNs appear more competitive than they do under settings that require learning transferable spatial structure from sustained novelty.

### 5.1.3 Compute profile

Fig. 8 compares inference latency, update cost, and parameter count. HEX-PMA remains competitive in model size and deployment-time inference, but incurs higher training-update cost due to nested attention.

---

[9]101 generation seeds per family, i.e. 202 maps in total.

This is the relevant trade-off: the method shifts cost toward training while remaining practical at control time, and it remains viable in regimes where flatter geometric attention encoders become too expensive.

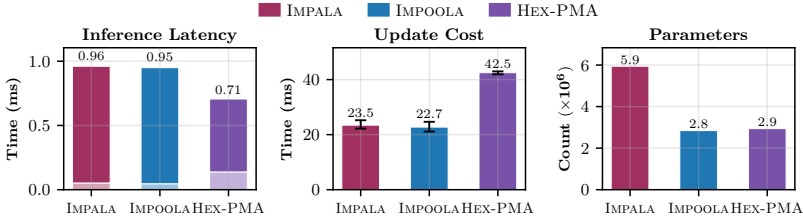

Figure 8: **Compute profile.** Inference latency, update cost, and parameter count; inference is split into observation construction and network forward pass.

### 5.1.4 Main Comparison Takeaway

The main comparison reveals a clear regime split: HEX-PMA is strongest under unbounded procedural training, while its advantage narrows under bounded training on a fixed finite map pool. This supports the view that the interface helps extract reusable spatial structure under sustained novelty, rather than merely improving fit to recurring layouts. The gains are obtained despite a bounded and lossy observation, whose limits are most visible in dense environments through active-cell selection and per-cell truncation (Tab. 3, App.).

## 5.2 Geometric encoder ablations

We next isolate differences among geometric encoders that operate on the same triangle-based observation. The purpose of this experiment is not to conduct an exhaustive survey of geometric encoders, but to test a narrower question: whether bounded hierarchical PMA aggregation sacrifices task quality relative to fuller attention-based alternatives in the regime where those alternatives remain computationally feasible. Accordingly, we compare HEX-PMA, HEX-ATTN, and a global attention baseline, denoted TRANSFORMER, while holding the geometric input fixed. These ablations are conducted in a simplified procedural regime with reduced primitive budgets. This reduction is necessary because, at the full primitive budgets used in the main experiments, HEX-ATTN already becomes memory-intensive, and the global attention baseline becomes impractical without additional modifications that would confound the architectural comparison. The ablation should therefore be interpreted as a controlled comparison: it asks whether hierarchical PMA remains competitive before flatter attention mechanisms become prohibitively expensive.

### 5.2.1 Performance under reduced budgets

In the reduced ablation regime, all three geometric encoders achieve closely matched success rates on both ID and OOD evaluation (Fig. 9). HEX-ATTN is marginally strongest in success, but the differences are small and within run-to-run variability. Computational scaling is shown in Fig. 10. Path efficiency shows a somewhat clearer distinction: HEX-PMA consistently outperforms HEX-ATTN on this metric and roughly matches the global Transformer. The main conclusion is, therefore, limited but important. In the reduced regime where all three encoders remain computationally feasible, replacing full self-attention with PMA does not produce a large task-quality penalty. The point is not that PMA uniformly dominates fuller attention, but that bounded hierarchical aggregation preserves most of the useful signal for control while avoiding the cost structure of flatter attention mechanisms.

### 5.2.2 Computational scaling

The compute profiles separate more clearly than the reduced-regime task performance. As primitive counts increase, global self-attention scales less favorably than the hierarchical hex-based variants, while compiled

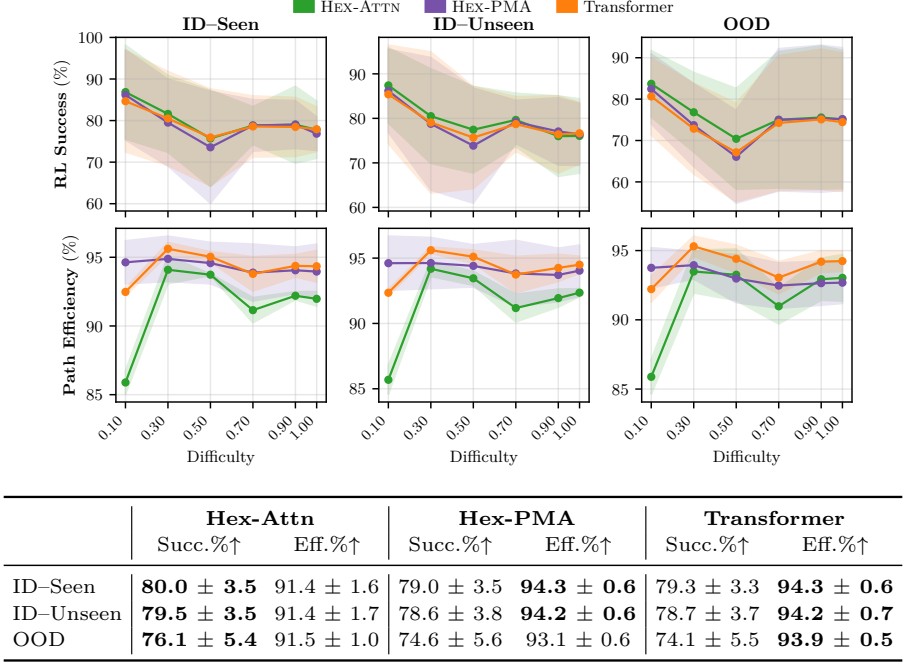

| | Hex-Attn | | Hex-PMA | | Transformer | |
|---|---|---|---|---|---|---|
| | Succ.%↑ | Eff.%↑ | Succ.%↑ | Eff.%↑ | Succ.%↑ | Eff.%↑ |
| ID–Seen | **80.0 ± 3.5** | 91.4 ± 1.6 | 79.0 ± 3.5 | **94.3 ± 0.6** | 79.3 ± 3.3 | **94.3 ± 0.6** |
| ID–Unseen | **79.5 ± 3.5** | 91.4 ± 1.7 | 78.6 ± 3.8 | **94.2 ± 0.6** | 78.7 ± 3.7 | **94.2 ± 0.7** |
| OOD | **76.1 ± 5.4** | 91.5 ± 1.0 | 74.6 ± 5.6 | 93.1 ± 0.6 | 74.1 ± 5.5 | **93.9 ± 0.5** |

Figure 9: **Geometric encoder ablations.** Success rate and path efficiency under reduced primitive budgets; table reports averages over difficulty.

HEX-PMA provides the most practical deployment profile (Fig. 10). Thus, full attention is viable in the reduced setting, but bounded hierarchical aggregation is the variant that continues to scale.

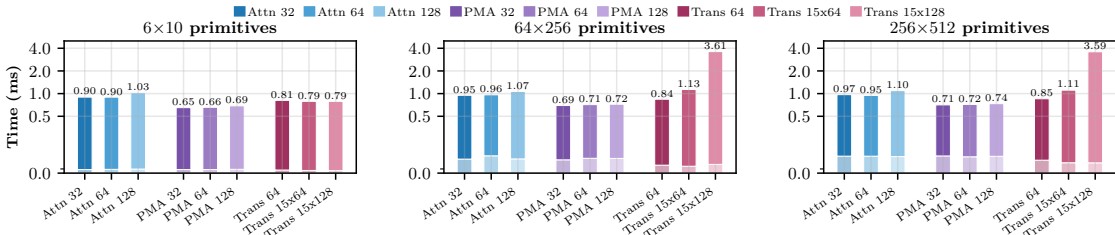

Figure 10: **Inference times of geometric encoder variants across primitive counts.**

### 5.2.3 Ablation takeaway

Taken together, the ablations show that bounded hierarchical geometric processing substantially improves computational scalability while preserving competitive task performance. In the reduced regime where all variants remain feasible, HEX-PMA shows no clear task-level disadvantage relative to HEX-ATTN or the global TRANSFORMER. As primitive counts increase, however, the flatter attention baselines scale much less favorably, whereas the hierarchical HEX-PMA design remains practical.

## 6    Discussion and Limitations

An interesting possibility suggested by the results is that hierarchical PMA helps extract task-relevant geometric glimpses from a bounded local observation. This interpretation is consistent with both the gains under procedural novelty and the saliency analysis in App. E.2, which suggests that the model concentrates on locally relevant geometric cues within the compressed interface. This is notable because the policy never

receives the full triangulation in unrestricted form. The observation function applies active-cell selection and per-cell capacity truncation, discarding primitives before encoding them (Tab. 3, App. D.3). Since truncation follows a fixed task-agnostic order, it does not preferentially preserve information that would favor the geometric interface. Masked PMA is therefore essential: it lets the encoder process variable-cardinality, padded sets without relying on tensor order or padding identity. The deployment-time truncation study in App. D.3 shows that performance degrades only slightly as per-cell capacity is reduced, suggesting that the interface tolerates moderate information loss. The main limitation, however, remains the assumption that structured geometry is available. This deliberately isolates the observation-interface question from perception, exploration, and state estimation, but also places the setup closer to fully observable map-based planning than to first-person embodied exploration. One might compare this setting to differentiable planning architectures such as Value Iteration Networks (Tamar et al., 2016); however, VIN-style methods implement explicit planning computations over a discretized grid MDP, whereas our approach keeps the planner generic and studies how a continuous geometric map should be exposed to an RL policy.

Generalization gains also remain uneven across held-out generators. Some structured OOD environments remain difficult for all methods, and the advantage of HEX-PMA is not uniform across generator families (per-generator results in Tab. 7, App.). This suggests that the learned PMA-based aggregation is not yet equally effective across the full range of geometric patterns considered here, and may transfer more reliably when test-time local structure remains closer to the regularities seen during training. A further limitation is that the geometric encoder ablation is conducted in a reduced procedural regime with smaller primitive budgets than those used in the main experiments. This is necessary to keep alternative attention-based geometric models computationally feasible under controlled conditions. The ablation should therefore be read as a controlled test of whether bounded hierarchical aggregation preserves performance before flatter attention mechanisms become prohibitively expensive, rather than as a broad benchmark of geometric encoders.

The hexagonal parcellation is an engineering choice enabled by the continuous geometric interface. Unlike raster observations, the partition need not inherit square-pixel discretization, and hexagons provide a convenient equal-area tiling with uniform six-neighbor connectivity and nearly isotropic local neighborhoods. This makes them a natural default for compact spatial aggregation. The choice is also compatible with biologically inspired spatial codes, such as the hexagonal firing structure of entorhinal grid cells (Moser et al., 2014), although the method itself only requires a consistent local spatial partition.

## 7  Conclusion and Future Work

We introduced a hierarchical geometry-first observation interface for sparse-reward spatial planning in RL and studied whether and when RL-based spatial planning benefits from it. The empirical results support two main conclusions. First, direct geometric structure is most valuable under sustained procedural novelty. In the unbounded training regime, HEX-PMA clearly outperforms CNN baselines, even though those baselines operate on occupancy rasters augmented with signed-distance-field channels. In the bounded regime, where agents repeatedly revisit a fixed finite map pool, this advantage becomes much smaller. This shows that the proposed interface primarily helps the agent learn reusable spatial structure, rather than merely fit recurring environments. Second, bounded hierarchical aggregation provides a practical route to geometry-first RL. In reduced-budget ablations, HEX-PMA remains competitive with fuller attention-based geometric encoders, while the scaling analysis shows that flatter attention mechanisms become substantially less practical as primitive counts increase. Thus, the proposed hierarchy is not only a representational choice, but also the mechanism that makes geometric observation tractable at the practical scales.

Although the present study assumes structured geometry, this setting is already relevant for applications where map-style information is available up front, such as planning over satellite or aerial imagery and related geo-AI domains. The natural next step is to relax this assumption and study partially observable settings, where the agent must act under occlusion, limited sensing range, or accumulated map uncertainty. Beyond observability, future work should enrich triangle primitives with semantic attributes, allowing the same interface to encode not only free-space geometry but also properties such as color, object type, danger, or other labels. A closely related direction is to combine the interface with perception or mapping systems that construct structured geometry online from noisy sensory input.

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

# A Context, Definitions, and Terminology

This appendix makes explicit the conceptual viewpoint that underlies the paper and clarifies the terminology used throughout. Its purpose is not to introduce an additional empirical claim beyond the main text, but to state clearly the abstraction-layer perspective from which the work is formulated.

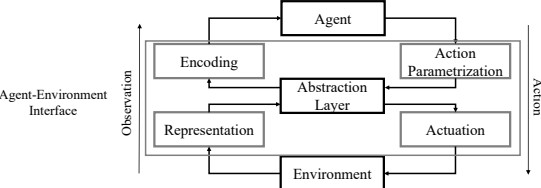

Figure 11: The geometry abstraction layer. Triangulated environment geometry serves as the structured intermediate representation between the raw environment and the RL observation encoder.

**Geometry as an abstraction layer.** In this work, we treat geometry as one possible abstraction layer for reinforcement learning in spatial decision-making problems. By this we mean that the agent interacts with a structured description expressed directly in geometric terms—such as positions, distances, orientations, boundaries, and occupancy—rather than through a rasterized image-like proxy.

**A decomposition of the agent–environment interface.** The agent–environment interface is defined by observation and action. On the observation side, we distinguish between *representation* and *encoding*. The *representation* specifies how relevant environment state is presented before learning, for example, as rasters or as bounded sets of geometric primitives. The deterministic, non-learned map from environment state to that representation is referred to as the *observation function*, and its codomain as the *observation space*. The *encoding* is the learned neural mapping that transforms this representation into the latent state representation used by the policy and critic. We use the term *observation interface* to denote the full pipeline formed by the observation construction together with the encoder that ingests it. On the action side, we distinguish between *parametrization* and *actuation*. The *parametrization* specifies the abstract form in which actions are predicted by the policy, while *actuation* concerns how those actions are realized in the underlying system. In the present work, the observation construction is deterministic and non-learned, while the encoder is part of the learned neural architecture of the RL agent. We acknowledge that the boundary between representation and encoding is not absolute in general, but for the purposes of this paper, the distinction is operationally clear and useful.

**Observation–action alignment as a motivating viewpoint.** A broader motivation for this work is the idea that spatial RL may benefit when observation and action are expressed in compatible abstraction layers. When observations are represented geometrically, it can be natural to formulate actions in the same spatial frame rather than tie them too closely to platform-specific realization details. We include this viewpoint only as a conceptual context for the present formulation, not as a claim established directly by the experiments. In robotics and planning-oriented systems, alignment between the abstraction level of observation and that of action is not unusual; it is often built into the separation between representation, planning, and execution. In reinforcement learning, however, this alignment is less often isolated and studied explicitly as an interface-design principle. A central conceptual aim of this work is to state that principle clearly and to examine one concrete realization of it on the observation side.

**Action as geometric completion of the same substrate.** Under this viewpoint, actions need not be treated as external to the abstraction layer in which the environment is represented; rather, they are elements of that same layer, which in the present work is instantiated by the observation space. In that sense, actions may be viewed as a form of geometric completion (*inpainting*) into the same structured substrate on which the environment is represented: the policy proposes geometric continuation or completion within the same spatial frame in which obstacles, free space, and goals are already expressed. We use this as a conceptual interpretation of the formulation rather than as a separate algorithmic claim.

**Scope of the present paper.** The experiments in the main text isolate the observation interface. They study a deterministic geometry-first observation construction together with a learned encoder, while keeping the downstream actor–critic architecture fixed. The action formulation used here is compatible with the same geometric frame, but the paper does not attempt a comparative study of alternative action abstractions. At the same time, the present line-segment action should be understood as a simple member of a broader family of geometric path parametrizations, with spline-based or higher-order variants obtainable through reparametrization.

### A.1 Hex Observation Construction

Figure 12 illustrates the egocentric construction. The lattice is centered on the agent, and the geometry is read out in the goal-aligned ego frame, whose $+x$ axis points along the agent–goal direction. The ego-to-world rotation is

$$\mathbf{R}_{\theta_t} = \begin{bmatrix} \cos\theta_t & -\sin\theta_t \\ \sin\theta_t & \cos\theta_t \end{bmatrix}, \qquad \theta_t = \text{atan2}(g_y - y_t, \ g_x - x_t). \tag{6}$$

This frame is *time-dependent*: the goal is fixed within an episode, but the agent position $\mathbf{x}_t$ evolves, so the heading $\theta_t$—and hence the center and orientation of the parcellation—are recomputed at every step. The observation is therefore a moving, goal-aligned window: each primitive is expressed in this frame, assigned to a cell, and aggregated.

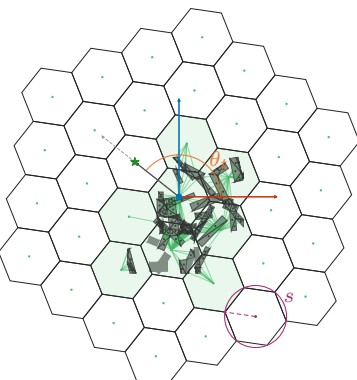

Figure 12: Hex-local observation: lattice of circumradius $s$ centered on the agent, heading $\theta_t$ aligning $+x$ with the goal. The gray dashed arrow marks the agent–goal axis, while the red and blue arrows are the world $x$ and $y$ axes, respectively. Active cells are marked in green, and active triangles are connected to the center with a green line.

# B    Neural Architecture

## B.1    Actor Critic Model

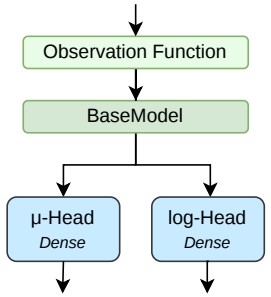

Figure 13: Actor network.

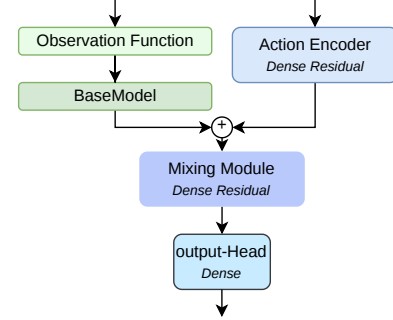

Figure 14: Critic network.

To isolate the effect of the observation interface from confounding architectural choices, we standardize all experiments around a shared modular Actor–Critic template. Each method differs only in the observation encoder, referred to as the *Base Model*, which maps its input representation into a common latent vector $\mathbf{h}_t \in \mathbb{R}^d$. This latent representation is then passed to the same policy head and twin critic heads for all methods (Figs. 13 and 14). The downstream decision-making architecture is therefore held fixed across the entire comparison.

## B.2    Baseline Details

This subsection specifies the baselines used in the main text.

**Raster observation.**    To preserve global context while limiting aliasing and information loss from a single discretization, raster baselines receive two map streams: a *local* map and a *global* map. Both maps use the same channel set. Since the environments are static, we precompute two geometry channels: a binary occupancy channel and a signed distance field (SDF) channel. The SDF channel stores the signed distance to the nearest obstacle boundary, providing the raster baselines with continuous metric information rather than only discrete occupancy. At each timestep, we additionally render dynamic channels for the agent position and the goal position. Thus, the raster baselines receive four channels in total: occupancy, SDF, agent position, and goal position. The global map $M_t^{\mathrm{glob}} \in \mathbb{R}^{G \times G \times C}$ is world-aligned and downsampled to a fixed resolution. The local map $M_t^{\mathrm{loc}} \in \mathbb{R}^{P \times P \times C}$ is an ego-centric crop centered at the agent and rotated so that the goal direction aligns with the positive $x$-axis. To express goal distance in a resolution-independent way, we normalize the ego–goal distance $L_t$ in the navigation vector by the crop circumradius $\rho_{\mathrm{crop}}$. Both streams and their channels are depicted in Fig. 15.

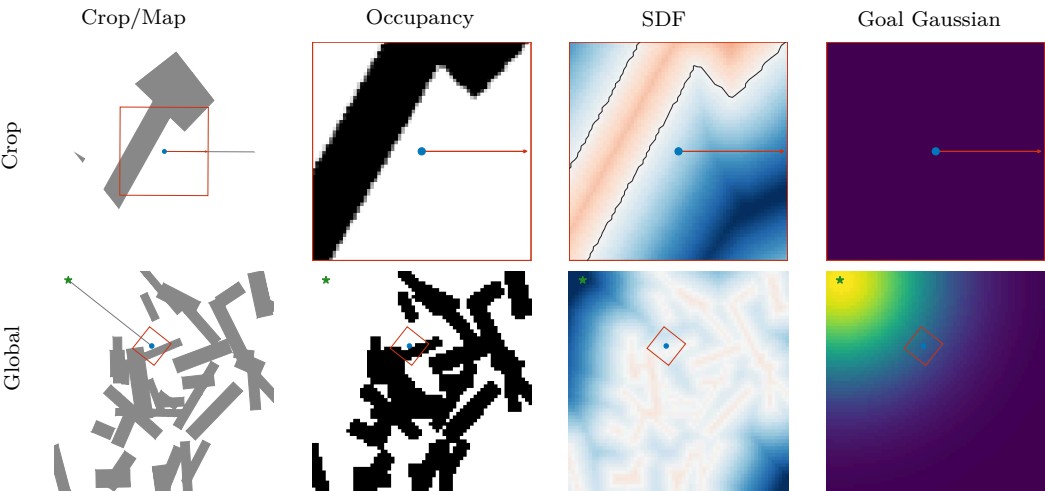

Figure 15: **Raster observation streams.** Rows: the ego-centric local crop (zoomed and rotated so the agent–goal axis points along $+x$) and the world-aligned global map. Columns: the per-stream channels. In the SDF channel, red marks the inside of obstacles and blue the surrounding free space; in the goal channel, yellow marks the goal location and fades to blue with distance. The agent-position one-hot channel is omitted as trivial.

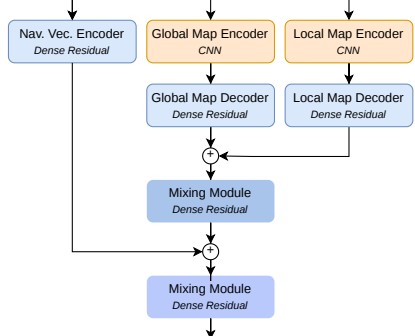

Figure 16: CNN encoder architecture for raster baselines (IMPALA/IMPOOLA).

**Raster model.** Each raster stream is encoded by its own image encoder. We evaluate the standard IMPALA-CNN (Espeholt et al., 2018), which flattens the final spatial dimensions, and IMPOOLA-CNN (Trumpp et al., 2025), which replaces spatial flattening with global average pooling (GAP) (Lin et al., 2014). We adopt only these convolutional encoders: unlike the original IMPALA agent (Espeholt et al., 2018), no recurrent (LSTM) state and no distributed training are used. All methods, raster and geometric, are memoryless and trained with the identical single-stream SAC backbone, so the baselines isolate the encoder architecture rather than the IMPALA learning system.

Let $f_{\mathrm{loc}}(M_t^{\mathrm{loc}})$ and $f_{\mathrm{glob}}(M_t^{\mathrm{glob}})$ denote the feature vectors produced by the local and global encoders, respectively. After projecting both streams to a width $d$, we combine the map features and then fuse them with the navigation vector $\mathbf{z}_t$:

$$\mathbf{h}_t = \mathrm{MLP}_{\mathrm{mix}_2}\Big(\mathrm{MLP}_{\mathrm{vec}}(\mathbf{z}_t) + \mathrm{MLP}_{\mathrm{mix}_1}\big(\mathrm{Dec}(f_{\mathrm{loc}}(M_t^{\mathrm{loc}})) + \mathrm{Dec}(f_{\mathrm{glob}}(M_t^{\mathrm{glob}}))\big)\Big). \qquad (7)$$

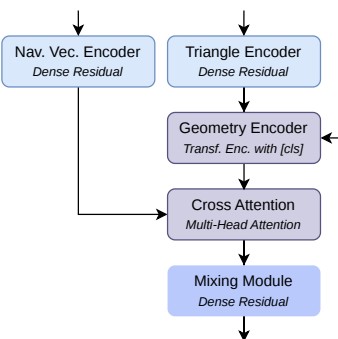

Figure 17: Global Transformer encoder architecture.

**Global Transformer.** The Transformer baseline (Fig. 17) mirrors the overall role of the geometric encoder while omitting spatial parcellation. Instead of first grouping primitives into local hex cells, it operates directly on the padded dense primitive set. Each primitive is embedded with a residual MLP, and the resulting sequence is processed by a standard Transformer encoder (Vaswani et al., 2017). Global information is represented through a CLS token (Devlin et al., 2019), which serves as a summary feature for the full triangulation. As in HEX-PMA, we then use cross-attention to extract task-relevant information from this global summary.

## C    Experimental Protocol

We summarize the environment distributions, training regimes, hyperparameters, metrics, and profiling protocol used throughout the experiments.

### C.1    Environment Distributions

**Map generators.**    At the beginning of each episode, a world map $G$ is sampled from a family of procedural generators $\text{Gen}_\psi : \mathcal{Z} \to \mathcal{G}$ using a random seed $\xi \in \mathcal{Z}$, i.e., $G = \text{Gen}_\psi(\xi)$. Once instantiated, the corresponding triangulated geometric representation $\mathcal{T}(G)$ is constructed once and reused throughout the episode. This triangulation forms the common geometric substrate on which all observation functions operate. The choice of generator family determines the qualitative structure of the environment, including obstacle layout, clutter type, and topological variation. In addition to this qualitative diversity, we explicitly control geometric complexity through a primitive budget. Unless otherwise stated, *complexity* refers to this budget.

Although the observation in the main method is defined over triangle primitives, procedural map generation is more naturally specified at the polygon level. We therefore parameterize environment complexity as a tuple in which the first entry denotes the maximum number of obstacle polygons and the second denotes the maximum number of triangles per polygon. Thus, a setting of $6 \times 12$ denotes an environment with up to 6 polygons, each represented by up to 12 triangles. This budget induces a first truncation level at the environment representation itself: generated maps are clipped or padded to fixed polygon and per-polygon triangle limits before observation construction. The observation function may then impose a second, independent truncation level when packing the bounded observation tensors used by the policy. The former controls environment complexity; the latter controls observation capacity.

**Complexity control.**    To obtain a controlled quantitative axis of environment variation, we group maps into two primary complexity branches according to their primitive budget. We refer to scenarios with at most $6 \times 10$ primitives as *simple*, and to scenarios with up to $64 \times 256$ primitives as *complex*, where the first factor denotes the maximum number of polygons and the second the maximum number of triangles used to represent them. During map generation, small polygons may be heuristically merged into larger ones when this preserves geometric fidelity. These preprocessing steps do not affect the experimental substrate itself: all observation functions operate on the final triangulation, which serves as the common geometric ground truth across methods. This separation allows us to vary structural diversity through generator family and geometric density through primitive count independently.

**Reset and goal sampling.**    Given a fixed map instance $G$, the initial position and goal pair $(\mathbf{x}_0, \mathbf{g}) \in F(G) \times F(G)$ are sampled from a reset distribution over free space, where $F(G)$ denotes the traversable subset of the environment. This reset distribution is parameterized by a scalar *reset difficulty* variable $t_{\text{diff}} \in [0, 1]$, distinct from the environment time index $t$.

Sampling proceeds in two steps: a start is first drawn in free space, and the goal is then constrained to an annulus around it whose radii grow with difficulty. Difficulty acts through three coupled linear schedules that jointly widen this annulus and raise the probability $p_{\text{h}}$ of a *hard start*—one drawn away from the map center, toward the periphery. Writing $\text{lerp}(a, b, t) = a + (b - a)t$, the admissible start–goal distance ring $[d_{\text{min}}, d_{\text{max}}]$ and the hard-start probability are

$$d_{\text{min}}(t_{\text{diff}}) = \text{lerp}(\underline{d}, \overline{d}, t_{\text{diff}}), \tag{8}$$

$$d_{\text{max}}(t_{\text{diff}}) = \max\big(\text{lerp}(\underline{d} + \delta, D, t_{\text{diff}}), d_{\text{min}}(t_{\text{diff}}) + \varepsilon\big), \tag{9}$$

$$p_{\text{h}}(t_{\text{diff}}) = \text{clip}_{[0,1]}\big(\text{lerp}(\underline{p}, \overline{p}, t_{\text{diff}})\big), \tag{10}$$

where $\underline{d}$ and $\overline{d}$ are the inner ring radius at the lowest and highest difficulty, $\delta$ is the initial ring width (so that $d_{\text{max}} = \underline{d} + \delta$ at $t_{\text{diff}} = 0$), $D$ is the maximal outer radius, $\underline{p}$ and $\overline{p}$ bound the hard-start probability, and $\varepsilon$ keeps the ring non-degenerate. Increasing $t_{\text{diff}}$ therefore widens the feasible goal ring and biases the start toward the periphery at once.

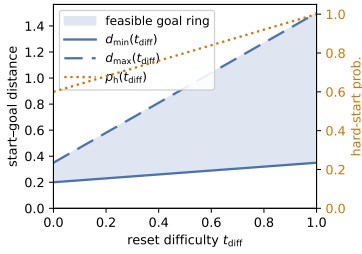

(a) Difficulty schedules.

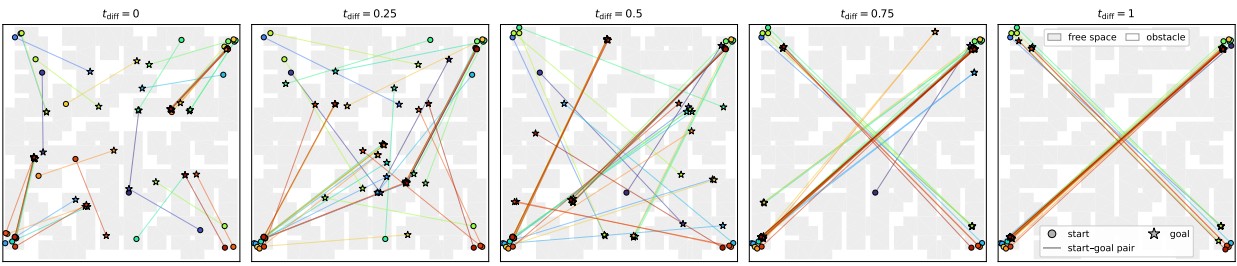

(b) Sampled resets at $t_{\mathrm{diff}} \in \{0, 0.25, 0.5, 0.75, 1\}$.

Figure 18: **Reset-difficulty parameterization.** (a) The distance ring $[d_{\min}, d_{\max}]$ and hard-start probability $p_{\mathrm{h}}$ vs. $t_{\mathrm{diff}}$. (b) Resets on a fixed map (one hue per episode, start ○ joined to goal ⋆): segments lengthen and starts move to the periphery as $t_{\mathrm{diff}}$ grows.

**Evaluation distributions.** Within each training regime, we evaluate all methods on three distributions.

**ID–Seen** corresponds to replay over maps generated from the same generator family and the same seed set encountered during training. In the bounded regime, this reduces to replay over the fixed training pool. In the unbounded regime, it corresponds to replay over maps that were encountered at least once during the expanding procedural curriculum.

**ID–Unseen** uses the same generator family as training, but a disjoint seed set that is never used during optimization. This evaluates in-family generalization to novel instances drawn from the same underlying distribution.

**OOD** evaluation is performed on generator families that are never used during training. This is our strongest test of transfer, as it assesses whether the learned representation remains effective on qualitatively different environment families generated by a disjoint set of procedural generators.

## C.2 Generator Statistics

## C.3 Training Details

**Training regimes.** We consider two complementary training regimes.

In the *bounded regime*, training is performed on a fixed finite pool of procedural maps defined by a seed set $\Sigma_{\mathrm{train}}$. The agent repeatedly revisits this same environment support under varying reset conditions. This regime measures how effectively a representation can exploit limited structural support and how well it extrapolates beyond a finite set of encountered layouts.

In the *unbounded regime*, training is performed on a continuously expanding procedural stream. At every reset, the agent is exposed to a new map instance. In this setting, performance depends on whether the agent can leverage increasing structural diversity and convert it into reusable spatial competence.

Together, the bounded and unbounded regimes distinguish learning under finite support from learning under sustained procedural novelty. This distinction is important because finite-support training may reward mem-

| Geometry statistics | | | | | Per-cell primitive density (percentiles) | | | | | |
|---|---|---|---|---|---|---|---|---|---|---|
| Generator | Example | Obs. Triangles | Free Triangles | Obs. Polygons | p50 | p90 | p95 | p99 | max | Exceeds $K_{cell}$ |
| forest |  | $109.3 \pm 19.6$ | $109.7 \pm 37.9$ | $8.9 \pm 3.3$ | 27 | 37 | 40 | 50 | 66 | ✓ (+3%) |
| ruins |  | $183.5 \pm 21.0$ | $200.8 \pm 23.9$ | $13.7 \pm 3.1$ | 50 | 73 | 82 | 96 | 123 | ✓ (+92%) |
| Hex-Forest |  | $51.0 \pm 14.0$ | $85.5 \pm 20.7$ | $9.5 \pm 2.3$ | 16 | 23 | 26 | 32 | 40 | – |
| dungeon |  | $123.7 \pm 10.4$ | $69.1 \pm 8.2$ | $4.5 \pm 1.5$ | 20 | 24 | 25 | 28 | 35 | – |
| city |  | $125.2 \pm 4.8$ | $186.4 \pm 8.5$ | $11.6 \pm 1.3$ | 25 | 28 | 29 | 31 | 34 | – |
| office |  | $341.6 \pm 12.0$ | $364.2 \pm 13.0$ | $17.4 \pm 3.4$ | 54 | 59 | 61 | 64 | 71 | ✓ (+11%) |
| maze |  | $235.3 \pm 2.9$ | $155.4 \pm 5.9$ | $36.0 \pm 0.0$ | 40 | 42 | 43 | 44 | 46 | – |
| teotihuacan |  | $707.4 \pm 198.7$ | $498.5 \pm 31.4$ | $38.7 \pm 5.3$ | 201 | 341 | 395 | 519 | 753 | ✓ (+1077%) |

Table 3: Generator statistics and per-cell truncation analysis. **Left:** Example observation and mean primitive counts (obstacle triangles, free-space triangles, obstacle polygons) across 100 sampled maps. **Right:** Distribution of per-cell primitive counts (p50–p99 denote percentiles; max is the observed maximum), pooled over all occupied cells; pXX is the count below which $XX\%$ of cells fall, so p50 is the typical cell and higher percentiles the busier ones. The final column indicates whether the maximum exceeds the cell capacity $K_{cell} = 64$, the per-cell budget used in all main experiments (together with $M = 15$ active cells per observation), with the percentage overflow in parentheses. This overflow is relative to the densest cell, $(\max - K_{cell})/K_{cell}$; e.g. +1077% for Teotihuacán means its busiest cell holds nearly 12× the capacity. Generators marked ✓ experience truncation under this default capacity budget. For the training generators (*Forest, Ruins*) the median cell stays within $K_{cell}$ and only the densest cells are truncated, whereas Teotihuacán (OOD) is the extreme case where truncation is pervasive yet still navigable (App. D.3).

orization or support exploitation, whereas unbounded training more directly tests whether a representation can accumulate reusable structure from continually changing environments.

**Random seeds.** All experiments are repeated over 5 independent random seeds. We report the mean and standard deviation across seeds for all quantitative results.

**Probabilistic adaptive curriculum.** Training long-horizon navigation policies from scratch on complex maps is highly inefficient. To address this, we employ a performance-driven adaptive curriculum that gradually increases task difficulty in response to the agent's competence. Specifically, we maintain an ex-

ponential moving average of recent success and adjust the mean of the reset-difficulty distribution so as to keep performance near a desired target level.

At each reset, $t_{\text{diff}}$ is drawn from a boundary-augmented truncated Gaussian about the curriculum mean $\mu_k$,

$$t_{\text{diff}} \sim (1-q_0-q_1)\, \mathcal{N}_{[0,1]}(\mu_k, \sigma^2) + q_0\, \delta_0 + q_1\, \delta_1, \tag{11}$$

where the point masses $\delta_0, \delta_1$ ($q_0=q_1=0.10$) preserve coverage of the trivial and hardest regimes and $\mu_k$ tracks recent success toward a 0.70 target via EMA control ($\sigma=0.04$).

**Discount scheduling.** In addition to difficulty adaptation, we employ an increasing discount-factor schedule during training. Early in training, a smaller discount biases optimization toward shorter-horizon behavior and stabilizes learning on locally solvable tasks. As training progresses, the discount factor is increased so that optimization gradually emphasizes longer-horizon planning and delayed rewards. In the main experiments, we use a linear schedule from $\gamma_{\text{start}} = 0.975$ to $\gamma_{\text{end}} = 0.99$ over the first $2 \times 10^6$ environment steps.

**Hyperparameters.** All methods share the same RL and training hyperparameters; only the encoder architectures differ.

| *SAC* | | *Training* | | *Discount schedule* | |
|---|---|---|---|---|---|
| Actor LR | $2 \times 10^{-4}$ | Replay buffer | 200,000 | Type | Linear |
| Critic LR | $2 \times 10^{-4}$ | Batch size | 64 | $\gamma_{\text{start}}$ | 0.975 |
| Entropy $\alpha$ | $1 \times 10^{-4}$ | Warmup steps | 10,000 | $\gamma_{\text{end}}$ | 0.99 |
| Target entropy | 0.03 | Updates/step | 1 | Schedule steps | $2\times10^6$ |
| Soft-update $\tau$ | 0.005 | Update freq | 1 | | |
| Policy delay | 1 | Max env steps | $2\times10^6$ | | |
| Actor freeze | 0 | | | | |

Table 4: Shared RL and training hyperparameters (all methods).

| Parameter | Hex-PMA | Hex-Attn | Transformer | Impala | Impoola |
|---|---|---|---|---|---|
| *Primitive / Image Encoder* | | | | | |
| Triangle MLP dim | 256 | 256 | $128 \rightarrow 64$ | – | – |
| Triangle MLP layers | 4 | 4 | 2+1 | – | – |
| CNN channels | – | – | – | [16,32,32] | [16,32,32] |
| Residual blocks / stage | – | – | – | 2 | 2 |
| CNN head dim | – | – | – | 256 | 256 |
| Spatial pooling | PMA | CLS + Self-Attn | CLS + Self-Attn | Flatten | GAP |
| *Set / Sequence Encoder* | | | | | |
| $d_{\mathrm{model}}$ | 64 | 64 | 64 | – | – |
| Intra-cell self-attn layers | – | 2 | – | – | – |
| Grid-level self-attn layers | – | 2 | – | – | – |
| Global self-attn layers | – | – | 4 | – | – |
| Intra-cell attn heads | 16 | 16 | – | – | – |
| Grid-level attn heads | 4 | 4 | – | – | – |
| Global self-attn heads | – | – | 8 | – | – |
| Cross-attn heads | 8 | 8 | 8 | – | – |
| PMA seeds $S_{\mathrm{tri}}$ | 16 | – | – | – | – |
| PMA seeds $S_{\mathrm{grid}}$ | 8 | – | – | – | – |
| *Feature Fusion* | | | | | |
| Navigation enc. dim | 256 | 256 | $256 \rightarrow 64$ | 256 | 256 |
| Navigation enc. layers | 4 | 4 | 4+2 | 4 | 4 |
| Map decoder layers | – | – | – | 1 | 1 |
| Map mixing layers | – | – | – | 2 | 2 |
| Output mixing layers | 4 | 4 | 4 | 2 | 2 |
| Output mixing dim | 256 | 256 | 256 | 256 | 256 |
| *Actor–Critic MLP* | | | | | |
| Critic action encoder | $2 \times 256$ | $2 \times 256$ | $2 \times 256$ | $2 \times 256$ | $2 \times 256$ |
| Critic mixing | $2 \times 256$ | $2 \times 256$ | $2 \times 256$ | $2 \times 256$ | $2 \times 256$ |
| Activation | SiLU | SiLU | SiLU[†] | SiLU | SiLU |
| LayerNorm | ✓ | ✓ | ✓ | ✓ | ✓ |

[†] Transformer critic uses ReLU; all other sub-networks use SiLU.

Table 5: Network architecture hyperparameters. Hex-PMA and Hex-PMA$_{\mathrm{mini}}$ (the reduced-budget variant used in the geometric encoder ablations, Sec. 5.2) share the same network architecture and differ only in the observation primitive budget ($K_{\mathrm{cell}}$=64 vs. 32).

## C.4 Metrics

**Training progression and sample efficiency.** Standard success-rate learning curves are not an informative proxy for sample efficiency in our setting, because training is governed by an adaptive difficulty controller that explicitly aims to maintain performance near a target success level. As a result, the observed success curve is intentionally flattened and does not faithfully reflect learning progress.

Instead, we quantify sample efficiency through progression through the curriculum. Our primary training-dynamics metric is the area under the curve (AUC) of the curriculum difficulty variable over environment steps. A method that reaches and sustains higher difficulty levels earlier in training is considered more sample-efficient. We additionally report asymptotic training success and the final attained curriculum level. Throughout, *asymptotic* denotes the mean of a training curve over the final 10% of environment steps, and *curriculum mean* denotes the controller mean $\mu_k$ of the adaptive reset-difficulty distribution (Eq. 11); the *asymptotic curriculum mean* reported in Tabs. 1 and 2 is therefore the average difficulty level sustained at the end of training.

**Final navigation performance.** To evaluate the spatial navigation capabilities of fully trained policies, we report navigation success rate and a path-efficiency measure relative to an oracle planner. Success rate captures whether the goal is reached within the episode budget, while path efficiency measures how economically the agent moves relative to the corresponding shortest feasible path in the environment. Path efficiency is computed over successful episodes only.

**Oracle planner.** The oracle planner computes shortest feasible paths using A* on a visibility graph built over the convex corners (extremity vertices) of the obstacle polygons, with Euclidean distance as both edge weight and heuristic. This yields the exact shortest Euclidean path through the free space of a 2D polygonal

environment. We use the `extremitypathfinder` package[10], which constructs the visibility graph via line-of-sight checks between polygon vertices and solves the search with `networkx.astar_path`. When numerical instability is detected (e.g., degenerate geometry or precision failures), the planner falls back to a discrete A* on a uniform grid derived from the same map. Both variants use Euclidean cost and differ only in the spatial representation used for path search. In our setting, path efficiency is particularly informative because successful behavior in highly non-convex environments is meaningful only when the executed trajectory also reflects effective long-horizon planning. Evaluating these quantities across the fixed difficulty grid provides additional insight into how performance degrades as the planning horizon increases.

**Deployment and computational metrics.** Finally, because one of our goals is to assess the practical viability of continuous geometric encoders for embodied systems, we report computational metrics. These include inference cost, update cost, and model size. Such measurements complement task performance by indicating whether the gains of a representation remain compatible with real-time deployment constraints.

## C.5 Profiling Protocol

Computational measurements are reported for inference latency, update cost, and parameter count. For inference, we separately profile (i) observation construction time and (ii) network forward-pass time. While these components are measured independently for analysis, the inference times reported in the main text correspond to their sum, which reflects the total end-to-end latency of one control step. Update cost is measured as the wall-clock time of a full training update, including all forward and backward passes required by the learning algorithm. Parameter count reports the total number of learnable parameters in the policy and value networks.

These measurements are intended to support relative comparison across methods under matched experimental conditions rather than to provide absolute hardware-independent timing claims.

## C.6 Hardware Details

| Component | Specification |
|---|---|
| CPU | AMD Ryzen Threadripper PRO 7985WX (128 threads) |
| GPU | 4× NVIDIA GeForce RTX 4090 (24 GB each) |
| Memory (RAM) | 512 GiB |
| Operating System | Ubuntu 22.04.5 LTS |
| Kernel Version | Linux 6.8.0-94-generic |

Table 6: Hardware and Software Configuration

---

[10]https://github.com/jannikmi/extremitypathfinder

# D    Extended Quantitative Results

## D.1    Per-Generator Evaluation Breakdown

|  | Impala | | Impoola | | Hex-PMA | |
|---|---|---|---|---|---|---|
|  | Succ.%↑ | Eff.%↑ | Succ.%↑ | Eff.%↑ | Succ.%↑ | Eff.%↑ |
| **ID−Seen** | | | | | | |
| **forest** | 71.5 ± 7.1 | **94.6 ± 0.3** | 81.7 ± 4.1 | 91.8 ± 0.3 | **86.5 ± 2.6** | 94.4 ± 0.1 |
| **ruins** | 41.4 ± 10.4 | 90.4 ± 0.6 | 52.0 ± 8.9 | 88.5 ± 0.5 | **67.4 ± 4.5** | **92.5 ± 0.3** |
| **ID−Unseen** | | | | | | |
| **forest** | 71.7 ± 7.5 | **94.7 ± 0.5** | 82.2 ± 5.0 | 91.9 ± 0.4 | **87.2 ± 2.6** | 94.5 ± 0.1 |
| **ruins** | 41.7 ± 12.1 | 90.2 ± 0.6 | 51.8 ± 10.1 | 88.4 ± 0.6 | **67.1 ± 5.9** | **92.4 ± 0.2** |
| **OOD** | | | | | | |
| **city** | 4.9 ± 6.2 | 85.8 ± 1.2 | 14.2 ± 9.8 | 86.0 ± 1.2 | **16.0 ± 9.8** | **89.7 ± 0.7** |
| **dungeon** | 41.7 ± 13.3 | 90.4 ± 0.9 | 57.0 ± 9.4 | 89.8 ± 0.8 | **62.2 ± 8.2** | **93.8 ± 0.4** |
| **Hex-Forest** | 76.4 ± 8.1 | 93.1 ± 1.0 | 85.8 ± 5.0 | 89.7 ± 1.4 | **94.0 ± 1.2** | **93.9 ± 0.6** |
| **maze** | 10.9 ± 5.9 | 83.6 ± 2.1 | 20.9 ± 7.9 | 86.7 ± 2.0 | **25.7 ± 9.0** | **91.0 ± 0.7** |
| **office** | 5.4 ± 5.3 | 82.0 ± 2.1 | 11.8 ± 8.9 | 83.9 ± 1.6 | **25.6 ± 8.2** | **88.7 ± 0.5** |
| **teotihuacan** | 37.0 ± 7.9 | 89.3 ± 0.7 | 46.7 ± 6.8 | 87.4 ± 0.5 | **47.0 ± 6.9** | **91.9 ± 0.2** |

Table 7: Full evaluation breakdown for the main comparison, reported separately for each generator.

## D.2    Training Curves

We report the full training dynamics for both the unbounded and bounded main-comparison regimes. For each regime, we show episodic success rate and curriculum mean difficulty as a function of environment steps, together with the corresponding AUC and asymptotic summary statistics. The curves include all 5 independent seeds in their entirety; a subset of seeds ran beyond the nominal 2M-step budget for completeness. In all cases, the performance trend remains consistent past the 2M cutoff used for the main comparison, supporting the use of 2M steps as a sufficient comparison horizon and showing that the relative ordering of methods is stable.

### D.2.1    Unbounded regime

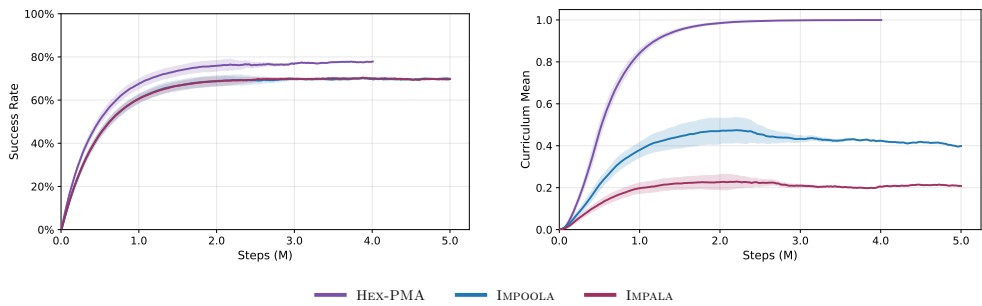

Figure 19: **Unbounded training curves (2M steps).** (*Left*) Episodic success rate. (*Right*) Curriculum mean difficulty. Shaded bands show mean ± std across 5 seeds. Some seeds continue beyond the nominal 2M-step comparison horizon.

### D.2.2 Bounded regime

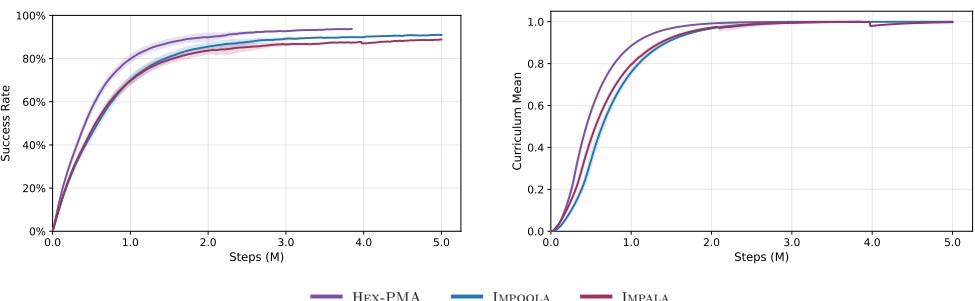

Figure 20: **Bounded training curves (2M steps).** (*Left*) Episodic success rate. (*Right*) Curriculum mean difficulty. Shaded bands show mean ± std across 5 seeds. Some seeds continue beyond the nominal 2M-step comparison horizon.

### D.3 Robustness to Deployment-Time Truncation

Finally, we examine the robustness of HEX-PMA to capacity constraints at deployment time. Starting from the trained model of the unbounded main experiment, we vary the maximum number of retained primitives per hex cell at inference time and evaluate the resulting policies without retraining. This directly probes sensitivity to primitive truncation and to capacity mismatch between training and deployment.

Fig. 21 shows the resulting performance across evaluation splits and difficulty levels. As expected, stronger truncation degrades both success and path efficiency. However, the effect is gradual rather than catastrophic, indicating that HEX-PMA tolerates moderate reductions in local geometric detail while retaining useful navigation behavior. This suggests that the model is not overly brittle to deployment-time observation caps, even though the representation relies on truncated local primitive sets. Fig. 22 visualizes the capacity-limited observation the policy receives along two successful trajectories, alongside the full triangulated state.

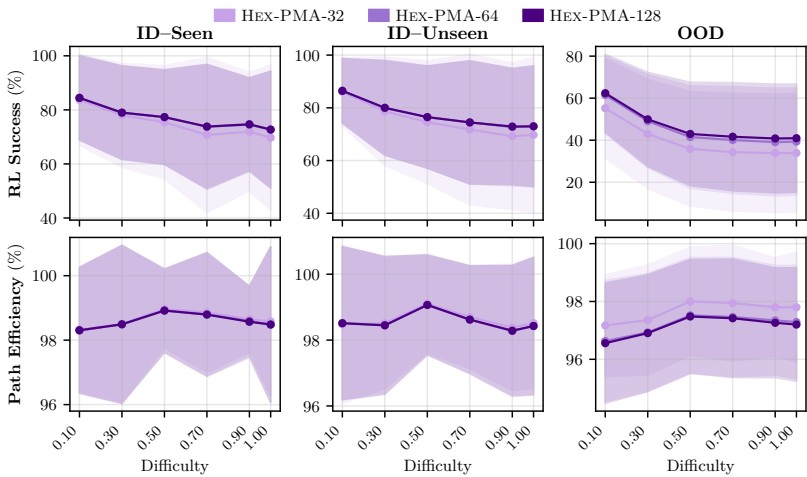

Figure 21: **Robustness of Hex-PMA to deployment-time truncation.** Performance of compiled and capacity-limited variants evaluated without retraining across ID–Seen, ID–Unseen, and OOD splits.

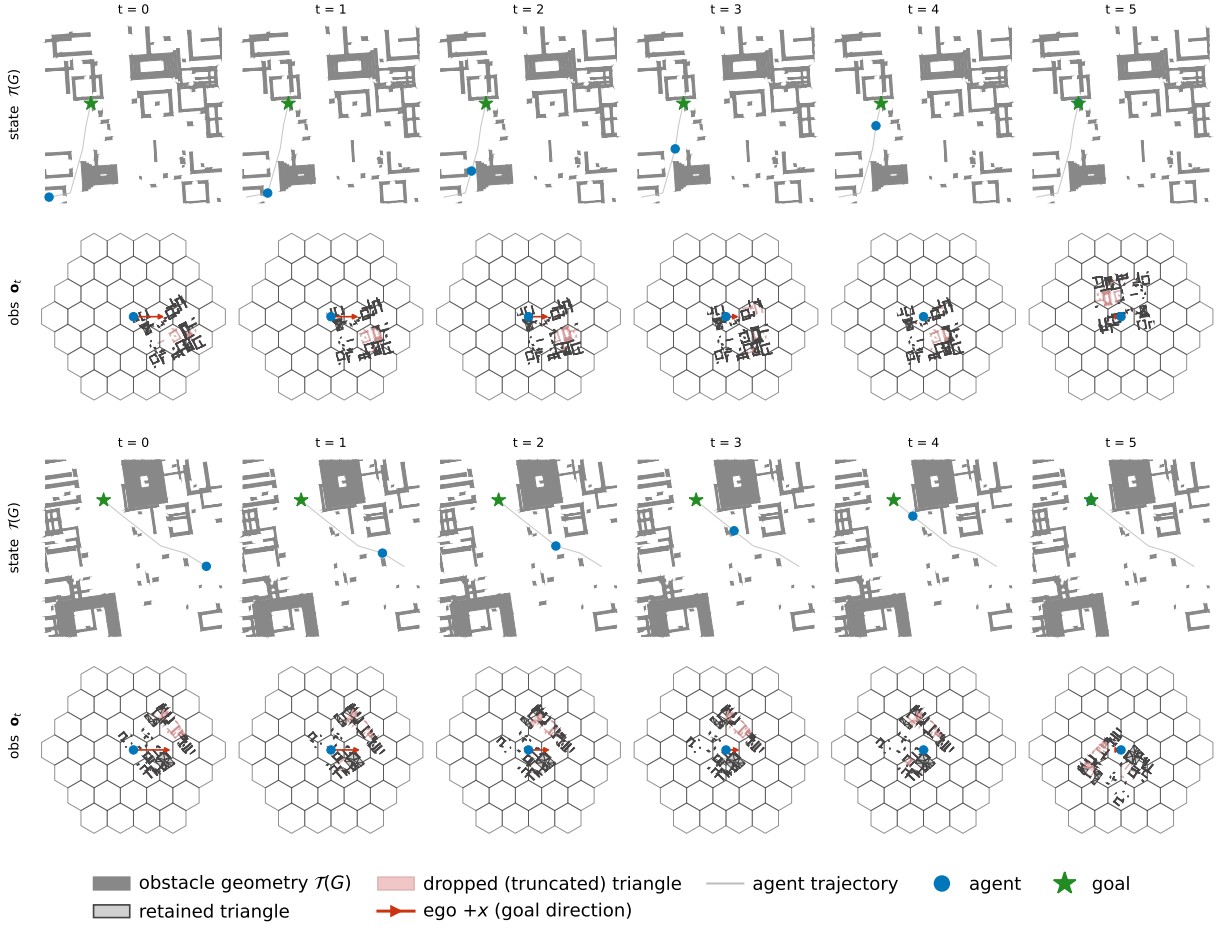

Figure 22: **Capacity-limited egocentric observation along successful trajectories (Teotihuacán).** Two successful agent rollouts (ep1, top; ep2, bottom); within each, the upper row is the true triangulated state $\mathcal{T}(G)$ at six steps and the lower row the observation the policy receives in the goal-aligned ego frame ($+x$ toward the goal). Markers and shading follow the legend: the agent reaches the goal along the grey trajectory; retained primitives are drawn solid (grey), and those that lie within range but are removed by capacity truncation are faded (red). Because the receptive field spans the whole map, the loss is a *capacity* effect ($K_{\mathrm{cell}}{=}64$ triangles per cell, $M{=}15$ active cells) rather than a limited range.

# E  Qualitative Analyses

## E.1  Qualitative Saliency Comparison

To better understand which parts of the observation influence each policy, we visualize saliency maps for representative states across the compared architectures. Although saliency does not by itself provide a complete explanation of model behavior, it provides a useful qualitative complement to the quantitative comparison. Fig. 23 shows matched saliency visualizations for Hex-PMA, Impala, and Impoola.

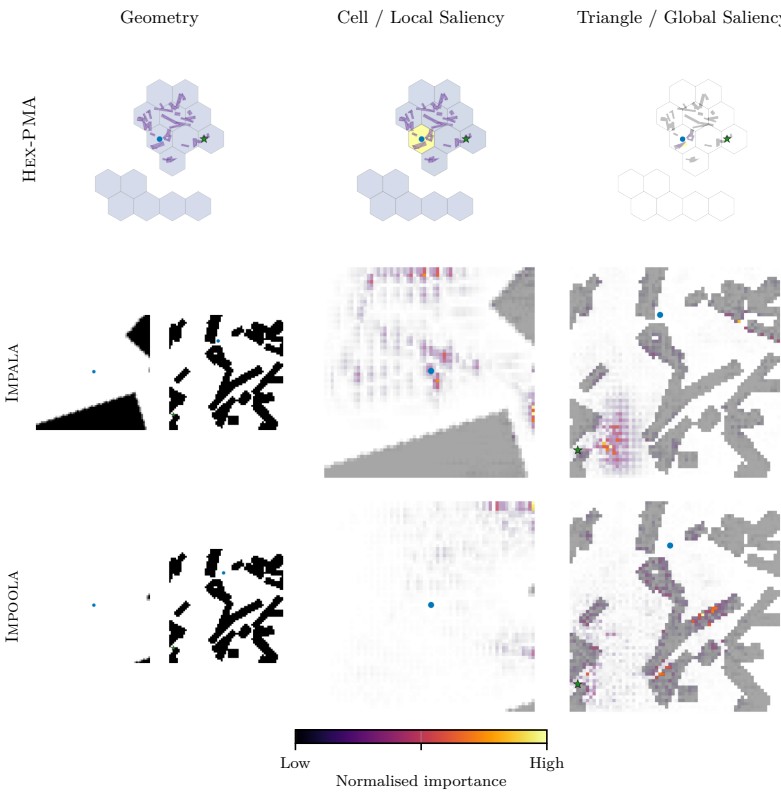

Figure 23: **Qualitative saliency comparison.** Attribution maps for Hex-PMA, Impala, and Impoola. The first column shows the input geometry. For Hex-PMA, the middle column shows importance aggregated per hex cell, and the right column shows per-triangle importance. For Impala and Impoola, saliency is overlaid on the ego-centric local crop and the global occupancy map. Colors follow the inferno scale from low (dark) to high (yellow). Agent: ●; goal: ★.

## E.2  Saliency Progression Analysis

The static snapshot in the qualitative saliency subsection above captures attribution at a single representative state. Here we study how saliency patterns evolve over the course of an episode across five independently sampled episodes per method, allowing assessment of cross-episode consistency.

**Saliency computation.**  For geometric methods (Hex-PMA, Hex-Attn, Transformer), saliency is computed as the gradient of the actor's log-probability with respect to the input features, aggregated either at the hex-cell level or at the individual triangle level. For raster methods (Impala, Impoola), vanilla gradient saliency is computed with respect to the local and global map pixel channels independently. All maps are normalized per frame to the $[0, 1]$ range and rendered on the inferno color scale (dark → low, yellow → high). The agent position is marked with a blue circle and the goal with a green star. This analysis follows the gradient-based attribution methodology popularized by Grad-CAM (Selvaraju et al., 2019).

Figures 24–28 display five independently sampled episodes per method, each spanning six evenly-spaced time steps.

Several consistent trends emerge across episodes. HEX-PMA (Fig. 24) reliably highlights the cells nearest to the optimal path, with the attribution front advancing toward the goal as the episode progresses. HEX-ATTN (Fig. 25) exhibits a similar directional bias but with somewhat broader cell-level attribution, consistent with its use of full intra-cell and grid-level self-attention rather than pooling-based aggregation. TRANSFORMER (Fig. 26) shows triangle-level saliency that is geometrically coherent but less spatially focused than the hex-partitioned methods, reflecting its lack of explicit local structure. IMPALA (Fig. 27) and IMPOOLA (Fig. 28) display saliency overlaid on both local and global map channels; the local crop consistently attracts the strongest gradients, with the global map contributing more diffuse, lower-magnitude attribution.

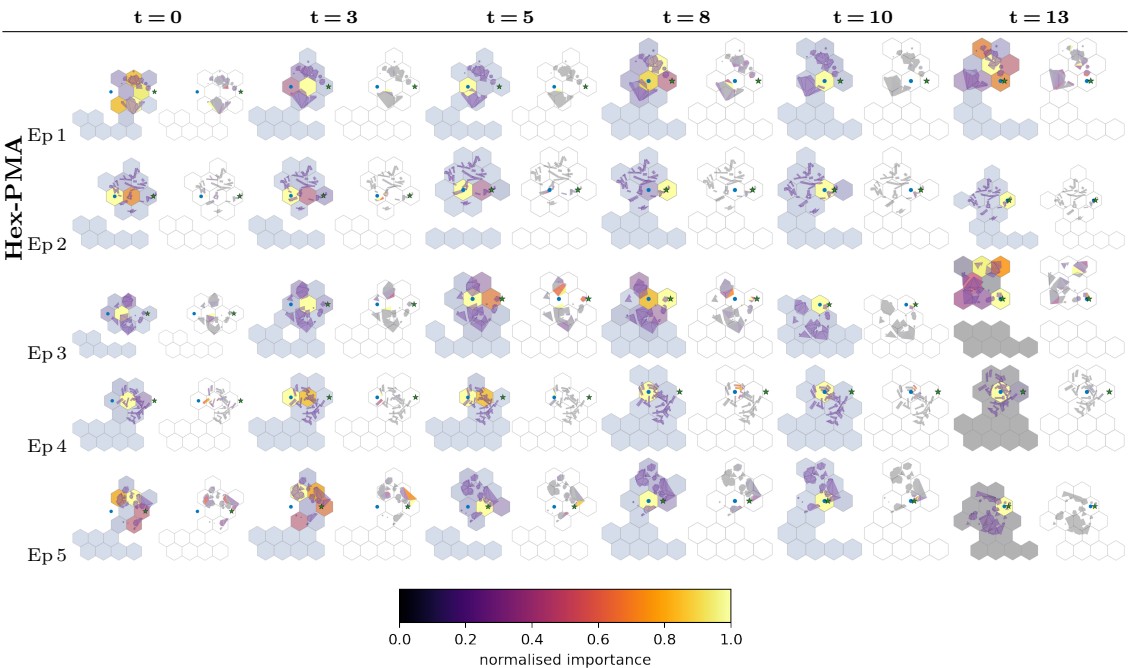

Figure 24: Saliency progression for HEX-PMA across 5 episodes. Cell (left) | Triangle (right). Columns show evenly sampled time steps within each episode.

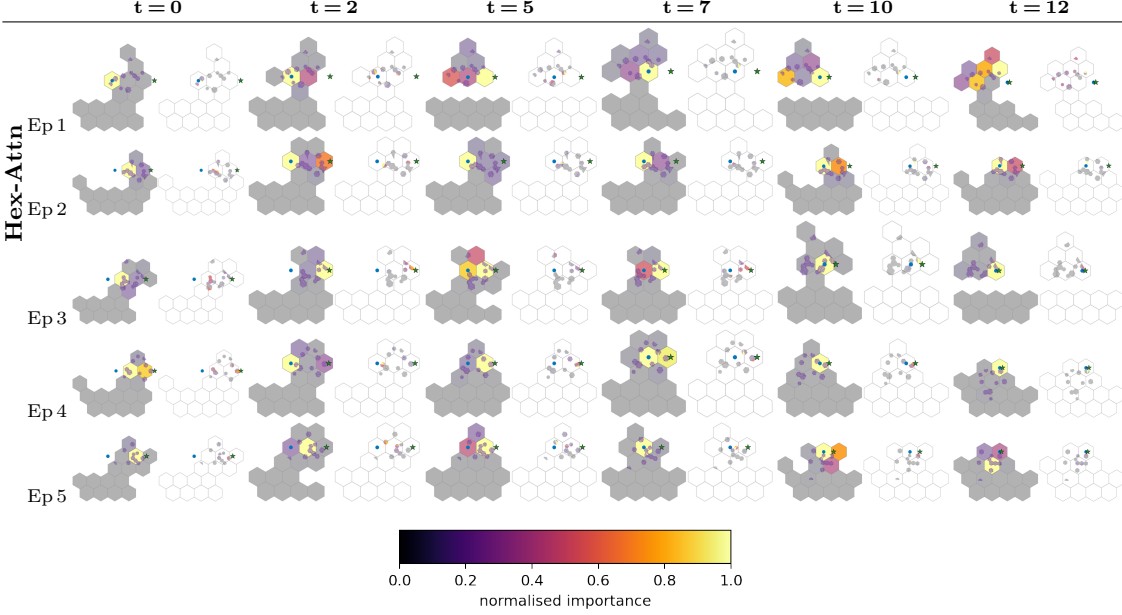

Figure 25: Saliency progression for HEX-ATTN across 5 episodes. Cell (left) | Triangle (right). Columns show evenly sampled time steps within each episode.

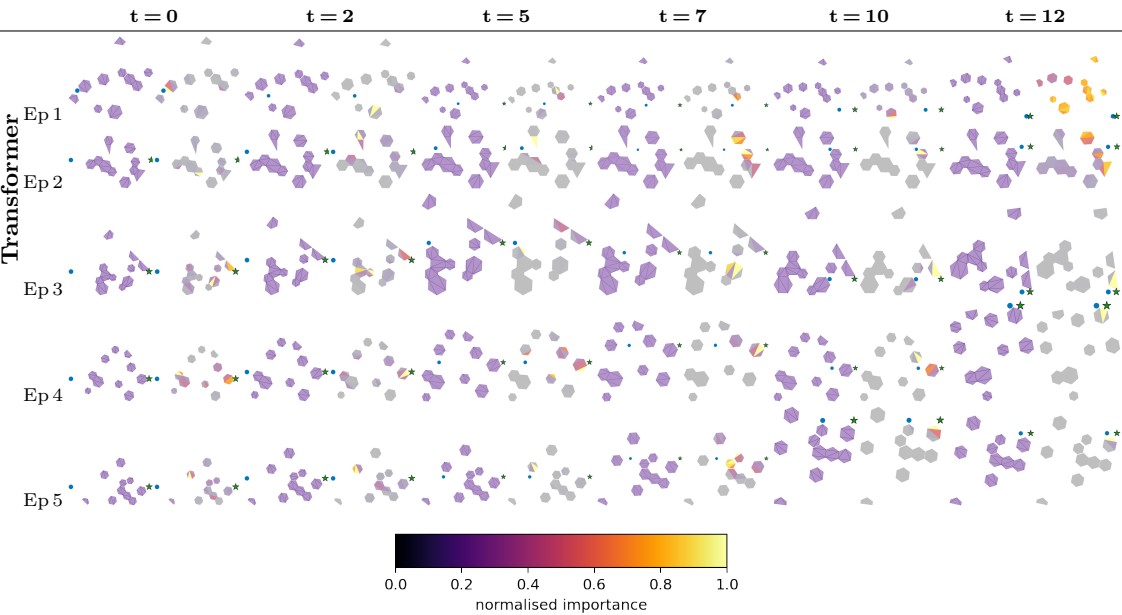

Figure 26: Saliency progression for TRANSFORMER across 5 episodes. Geometry (left) | Triangle saliency (right). Columns show evenly sampled time steps within each episode.

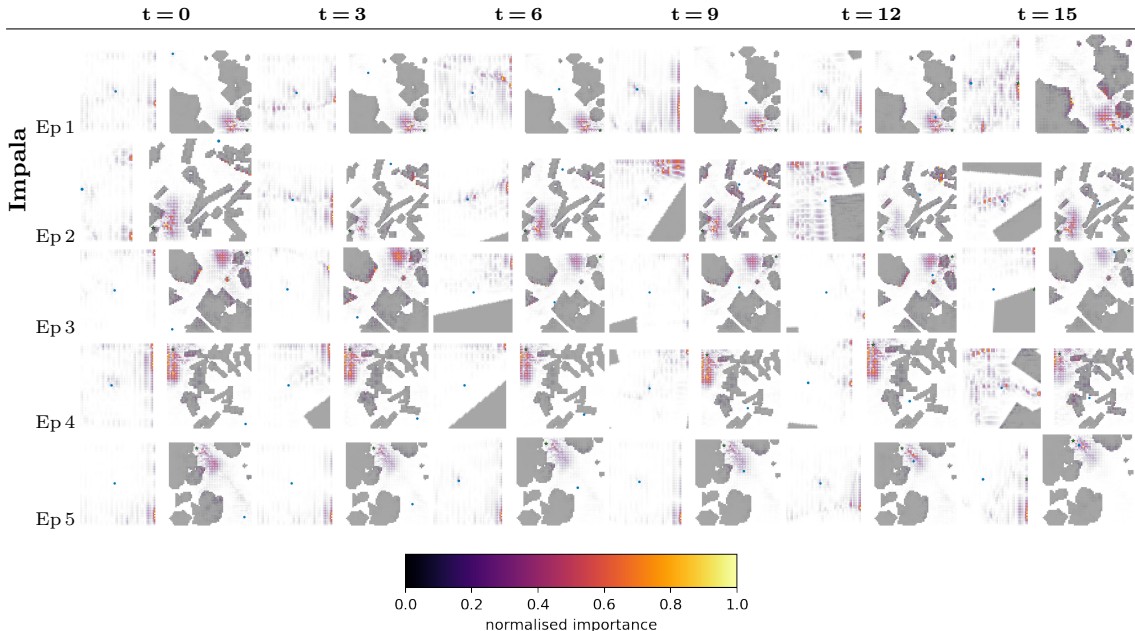

Figure 27: Saliency progression for IMPALA across 5 episodes. Local (left) | Global (right). Columns show evenly sampled time steps within each episode.

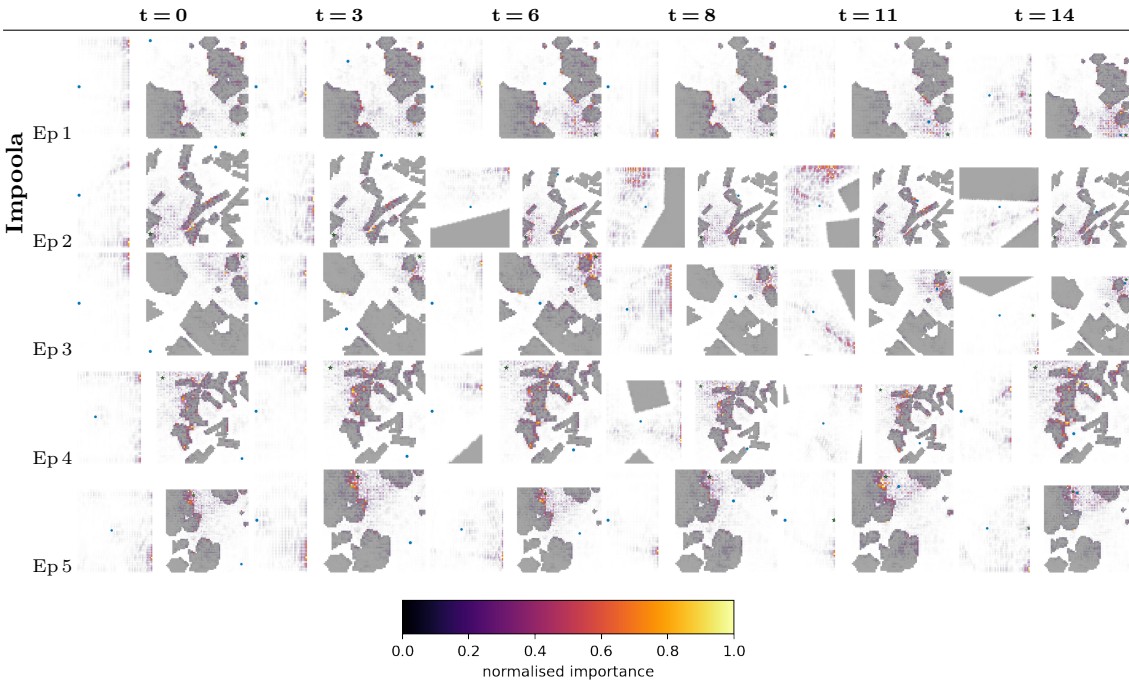

Figure 28: Saliency progression for IMPOOLA across 5 episodes. Local (left) | Global (right). Columns show evenly sampled time steps within each episode.

