# OpenReview forum: "A Hierarchical Geometric Observation Interface for Spatial Planning in Reinforcement Learning"
_TMLR — Under review for TMLR_

### Review · Reviewer_crBS · 2026-05-21

**Summary Of Contributions:**

The authors propose an off-the-shelf method that encodes the observation into a hexagonal grid and compared with IMPALA and IMPOOLA. The proposed method, Hex-PMA, shows superior performance in procedurally-generated 2D navigation tasks when the training data is unbounded. Furthermore, the proposed method generates a more efficient path in successful trajectories.

However, there are multiple weaknesses, especially the TMLR criteria (soundness, clarity). Please see the explanation in that section.

Other Weaknesses:

* In Section 5.1.2, the authors claim that Hex-PMA shows the strongest path efficiency in general. However, according to Figure 7, all results are within the confidence interval.

* The proposed methods require a full observation environment. I think the method can work in a POMDP scenario, although there is a computational overhead for the triangularization of newly observed obstacles.
* Furthermore, the authors only tested in static environments, which raises the question of how to apply it in dynamic environments.

**Additional Comments:**

Although it can be considered as concurrent work and a preprint, the work below can be relevant:

Sepúlveda, Carlos S., and Gonzalo A. Ruz. "Critic-Free Deep Reinforcement Learning for Maritime Coverage Path Planning on Irregular Hexagonal Grids." arXiv preprint arXiv:2603.28385 (2026).


Also, in terms of hexagonalization of the environment:

Kim, Jun-Ho, and Hanul Sung. "A Hexagon Sensor and A Layer-Based Conversion Method for Hexagon Clusters." Information 15.12 (2024): 747.

Duszak, Piotr. "SLAM on the Hexagonal Grid." Sensors 22.16 (2022): 6221.

**Audience:**

Yes

**Audience Explanation:**

If the authors show the potential in the POMDP scenario, the paper can be attended more easily. On the other hand, the authors might want to connect with the grid cell in neuroscience that maps a 2D environment into a hexagonal grid pattern. See the review paper:

Moser, Edvard I., et al. "Grid cells and cortical representation." Nature Reviews Neuroscience 15.7 (2014): 466-481.

**Broader Impact Concerns:**

I do not have an ethical concern regarding this work.

**Claims And Evidence:**

No

**Claims Explanation:**

Many important details are missing, making it difficult to evaluate the paper. Furthermore, some details are included in the appendix but are never referenced in the main text (e.g., "task difficulty" on page 18).

Please address the following issues and missing information:

* Observation Formation: Clarify the observation formation process, specifically regarding the SDF, position, and static environment features.
* Undefined Variables: The variable $s$ in Equation (1) is not defined. Furthermore, what does $R_t$ represent in the last sentence of page 4?
* Task Difficulty: On page 18, this is roughly defined as when to reset to $[0, 1]$, but the authors do not describe exactly how this is used to sample from the reset distribution.
* Terminology: Clarify the definition of "asymptotic curriculum mean." The appendix states it relates to curriculum difficulty, but the precise meanings of "asymptotic" and "curriculum" in this context are unclear.
* Baselines and Architecture:
1.  Why does the Transformer baseline not use hex spatial parcellation? I assume it substitutes the PMA in Figure 4 with a standard Transformer, but this must be explicitly described.
2. Why use a hexagonal shape? Please justify this choice or test other shapes (e.g., square, triangle).
3. The definition of "Raster" needs further explanation. Specifically, IMPALA's primary contribution is the distributed actor-critic framework, not the architecture itself. I suspect the authors followed the setting from Trumpp et al. (2025), which demonstrates the advantage of substituting flattening with global average pooling in IMPALA. This should be clarified.
4. Furthermore, the original IMPALA architecture uses an LSTM. Did the authors also use an LSTM here?
* Missing Details: Details regarding unbounded training are missing.
* Figures: Figures 16 and 17 lack legends and descriptions for each line.

**Requested Changes:**

* In footnote 7, Sec . => Sec.
* Figure 10 is too small. The table might be better.
* Regarding saliency visualization, it is good to cite related papers such as Grad-CAM.

Selvaraju, Ramprasaath R., et al. "Grad-cam: Visual explanations from deep networks via gradient-based localization." Proceedings of the IEEE international conference on computer vision. 2017.

---

> ### Author Response · Authors · 2026-06-14
>
> We thank the reviewer for the detailed, careful review; the checklist of missing items was especially helpful. All paper changes are marked in blue.
>
> **W1. Path efficiency within confidence intervals.** We have recalibrated: in the bounded regime (Sec. 5.1.2) the text now states that path-efficiency differences are small and fall within seed variability, and we no longer claim the strongest overall path efficiency there. In the unbounded regime we keep only the trend supported by the curves (raster baselines decline with difficulty, Hex-PMA stays nearly flat), and our primary claims rest on success rate, where the separation is large.
>
> **W2. POMDP applicability.** The aim of this paper is narrower than a POMDP solution: we set out to find a good observation interface for spatial RL, so we treat partial observability as future work rather than something we claim to solve here. That said, the setup is already partially observed in practice: active-cell selection and per-cell truncation give the policy a bounded, lossy, moving goal-aligned window (Sec. 3.2.1; Teotihuacán filmstrip, App. D.3), and the deployment-time truncation experiment (App. D.3) shows the encoder tolerates removed primitives without retraining. Extending to online, incrementally-observed geometry is the natural next step, stated as future work in the Conclusion.
>
> **W3. Dynamic environments.** Future work in the Conclusion; the per-step recomputation of the ego-frame observation (App. A.1) means nothing in the interface assumes a static scene, though we make no empirical claim.
>
> **Issues and missing information:**
>
> **(1) Observation formation.** App. B.2 now fully specifies the raster interface: both streams share four channels (occupancy; SDF; agent; goal), geometry precomputed per static map and agent/goal rendered per step; the global map is world-aligned and downsampled, the local map an ego-centric crop rotated to the goal. A new figure (App. B.2) depicts both streams and their channels.
>
> **(2) Undefined variables (Eq. 1, p. 4).** All symbols are now introduced (axial coordinates (q, r), circumradius s), and z_t is defined component-by-component (Sec. 3.2.1). With figures in the appendix.
>
> **(3) Task difficulty.** App. C.1 gives the full parameterization (three closed-form schedules with all constants, the two-step start/goal sampling, and a figure of schedules and sampled resets), referenced from the main text (Sec. 4.2).
>
> **(4) "Asymptotic curriculum mean."** Defined in App. C.4: "asymptotic" = mean over the final 10% of steps; "curriculum mean" = the controller mean mu_k of the adaptive reset-difficulty distribution; the controller itself is specified in the same appendix.
>
> **(5) Baselines and architecture.** The Transformer baseline is now described (Sec. 4.3, App. B.4): same triangulated geometry without hex parcellation, CLS-token aggregation, same fusion as Hex-PMA; a footnote notes Hex-Attn is this baseline plus hexagonal parcellation. The hexagonal choice is discussed as deliberate but not load-bearing (we do not ablate cell shape), with the grid-cell connection added (Moser et al. 2014). "Raster" means the IMPALA-CNN encoder (flattening) and Impoola (global average pooling; Trumpp et al. 2025), standard CNN encoders in deep RL, not the IMPALA distributed learner. No method uses recurrence (LSTM); all are memoryless on the identical single-stream SAC backbone (App. B.3).
>
> **(6) Unbounded training details.** App. C.2 specifies both regimes (unbounded = a new map every reset from an unbounded seed stream; bounded = a fixed 101-seed-per-family pool, round-robin), and App. C.3 the ID-Seen / ID-Unseen / OOD splits.
>
> **(7) Figures 16-17.** Legends and per-line descriptions added; the attribution method is documented (gradient-based, in the spirit of Grad-CAM; Selvaraju et al. 2017, now cited).
>
> **Requested changes.** "Sec ." to "Sec." fixed globally; Figure 10 enlarged (kept as a figure to match Figure 8); Grad-CAM cited (Selvaraju et al. 2017).
>
> **Related work.** Added the grid-cell review (Moser et al. 2014). The hexagonal-discretization works (Sepúlveda & Ruz 2026, concurrent; Kim & Sung 2024; Duszak 2022) are relevant context, but our contribution is at the observation-interface level, so we draw no direct methodological link.

---

> > ### Comment · Reviewer_crBS · 2026-06-16
> >
> > I appreciate the authors for improving the paper. I will carefully read the response and the manuscript as early as possible.

---

> > ### Comment · Reviewer_crBS · 2026-06-28
> >
> > Thank the authors again for their effort. The current version is clearer than the initial submission.
> >
> > Regarding Task Difficulty, could you please clarify whether the linear interpolation argument can represent task difficulty? It looks too naively defined, and depending on the environmental structure. For example, if the environment is a hotel that has a long corridor, then, t=0 and t=1 might not be that different, and if the environment is simple in one axis (say x-axis), while difficult in another (y-axis), depending on the axis, the actual difficulty might be different.

---

> > > ### Author Response · Authors · 2026-06-30
> > >
> > > We thank the reviewer for raising this point; we agree that the terminology deserves clarification.
> > >
> > > As we described in the Appendix, the parameter $t_{\mathrm{diff}}$ does not linearly interpolate between two concrete tasks. It parameterizes a stochastic reset distribution over feasible start–goal pairs: at each reset, the start is sampled in free space and the goal is sampled from an annulus whose radii depend on $t_{\mathrm{diff}}$. Thus, $t_{\mathrm{diff}}$ is not a direct measure of the realized difficulty of any individual episode, but a scalar curriculum variable controlling the expected planning horizon. In sparse-reward navigation, expected horizon is a meaningful proxy for difficulty, although the realized difficulty of a particular instance also depends on topology, bottlenecks, clearance, and directional structure.
> > >
> > > The hotel/corridor example illustrates this distinction. A generator may be directionally biased: distance along one axis may imply harder planning than the same distance along another. Our reset construction deliberately does not reweight or alter the generator distribution to remove such biases; it preserves the structural variation induced by the generator while controlling start–goal separation through the reset sampler. This is also why we train and evaluate across multiple procedural generators rather than relying on a single environment family.
> > >
> > > We use the term "difficulty" because $t_{\mathrm{diff}}$ orders the curriculum from easier to harder reset distributions, and expected planning horizon is a meaningful proxy for difficulty in sparse-reward navigation. We agree that the term should not be read as exact per-instance difficulty. If the reviewer feels that a term such as reset-distance would avoid ambiguity, we would be happy to adopt it.

---

### Review · Reviewer_GMXL · 2026-05-23

**Summary Of Contributions:**

To address the aliasing and information loss caused by discretizing continuous environments into raster grids, the authors propose a geometry-first observation interface for spatial RL. They introduce Hex-PMA, a hierarchical architecture operating directly on triangulated obstacle meshes. By combining hex-local spatial parcellation with multi-head attention pooling, it compresses variable-sized geometry into a fixed-length vector, elegantly bypassing the computational bottleneck of global attention. They evaluate this on sparse-reward point-navigation using a frozen SAC backbone, explicitly comparing bounded (finite map pool) versus unbounded (continually generated) training regimes.

Strengths:
- Smart architectural design: Hex-PMA effectively solves the $O(N^2)$ scaling bottleneck of applying flat attention to raw polygons, making geometric processing computationally tractable.
- Excellent experimental isolation: Freezing the SAC backbone cleanly separates the representation's impact from RL optimization confounds, which is notoriously difficult.
- Nuanced evaluation: The explicit contrast between bounded and unbounded training provides a refreshingly honest look at exactly when this geometric inductive bias is actually beneficial.

Weaknesses:
- Unrealistic perception assumptions: The method assumes a perfect, noise-free triangulated map. This completely bypasses the messy reality of noisy, incomplete meshes produced by real-world SLAM or depth-estimation pipelines.
- Limited scope of advantage: The performance gap over CNN baselines essentially vanishes in bounded training, restricting the method's utility to scenarios demanding sustained, continuous novelty.

**Audience:**

Yes

**Audience Explanation:**

Yes, the intersection of representation learning and embodied RL is a major focus for the TMLR audience. Showing that continuous geometric representations offer a distinct advantage specifically in preventing map-memorization under continual novelty is a valuable insight. Researchers working on generalization, Sim2Real transfer, and robotic navigation will find the architecture and the bounded/unbounded analysis highly relevant to their work.

**Broader Impact Concerns:**

I do not have any major ethical concerns regarding this submission that would require a detailed Broader Impact Statement. This is foundational, algorithmic work focusing on representation learning and computational efficiency within simulated RL environments. While improvements in spatial navigation algorithms theoretically dual-use implications for physical robotics (e.g., military drones or surveillance systems), this research is abstracted at the level of the observation interface. It does not introduce novel, specific capabilities or harms beyond those already inherent to the general field of autonomous path planning.

**Claims And Evidence:**

Yes

**Claims Explanation:**

Yes, the claims are well-supported by clear and convincing empirical evidence. I found the narrative to be refreshingly honest; the authors do not overclaim that their method universally beats CNNs. Instead, they hypothesize that geometric representations prevent the agent from exploiting fixed map geometries, and they prove this by showing the divergence in performance between bounded and unbounded training. The learning curves clearly demonstrate that while CNNs can memorize a finite training set, Hex-PMA actually forces the learning of generalizable spatial structures.
Furthermore, the claim regarding computational efficiency is backed up by solid ablation studies demonstrating that flat attention baselines fail to scale, whereas the hierarchical approach maintains a manageable footprint.

**Requested Changes:**

Strengthen: The "perfect map" assumption is the biggest gap in the paper. I would also love to see an experiment (or at least a highly detailed discussion) addressing the robustness of Hex-PMA to noise. I suggest simulating real-world perception by injecting noise into the triangulations during evaluation, e.g., randomly dropping triangles. We need to know if the architecture completely collapses under partial observability and noise.

Strengthen: I would love to see a deeper dive into why the CNNs catch up in the bounded regime. You provide some excellent saliency visualisations in the appendix; expanding on this in the main text to compare the feature representations or saliency maps of CNNs vs. Hex-PMA when trained on a fixed map pool would greatly enrich the analysis.

---

> ### Author Response · Authors · 2026-06-14
>
> We are glad the reviewer found the architecture and experimental design compelling, and we are grateful for the two concrete suggestions, both of which made the paper stronger. All paper changes are marked in blue.
>
> **R1. "Perfect map" assumption / robustness to dropped triangles.** An experiment of exactly this kind was already part of the original submission; the revision makes it more prominent and easier to interpret. The deployment-time truncation study (App. D.3) takes the trained unbounded-regime model and reduces the per-cell primitive capacity at inference time without retraining, dropping triangles from the observation in an order that is deterministic but effectively random with respect to size and task relevance. Performance degrades gradually rather than catastrophically across ID-Seen, ID-Unseen, and OOD: the architecture does not collapse when primitives are removed.
>
> We made this experiment easier to find and interpret:
> - it is now referenced from the main text (Sec. 3.2.2, Sec. 4.1, Discussion) rather than only the appendix;
> - we clarify that truncation is a designed property of the training data, not a corner case: the training families are chosen so that per-cell primitive counts exceed the observation capacity (Ruins substantially), so the policy already trains and acts on partial views;
> - we added visualizations: the Teotihuacán progression filmstrip (App. D.3), showing side by side the true triangulated state and the capacity-limited observation the policy receives along successful trajectories; an annotated hex observation-construction figure (App. A.1); and a raster-streams figure showing both streams and their channels (App. B.2).
>
> A full study with SLAM-like noise (vertex perturbation, spurious or missing geometry, drift) is the natural next step, which we now state as future work together with the partial-observability extension. The masked set-operator design is what makes these extensions possible without architectural changes.
>
> **R2. Why do CNNs catch up in the bounded regime?** We offer this as a hypothesis, not a result. A plausible explanation is memorization capacity: with a fixed pool of 101 maps per family revisited for millions of steps, a CNN has ample opportunity to memorize layouts, whereas the bounded, truncated PMA summaries are a poorer substrate for instance-specific detail — which would explain why the geometric interface's edge appears mainly under sustained novelty. The revised Sec. 5.1.2 states this as a possible reason with no direct support in the results; the saliency analysis (App.) is qualitative evidence only.
>
> **On the noted limitation "advantage vanishes in bounded training".** We read this as a finding rather than a flaw, and the revision frames it accordingly: the bounded regime shows the geometric interface's benefit is tied specifically to settings that require learning reusable spatial structure, not to better curve-fitting on a fixed support.

---

### Review · Reviewer_7CvC · 2026-06-01

**Summary Of Contributions:**

The paper presents a hierarchical, set-based geometric observation interface to encode spatial maps for reinforcement learning navigation. Unlike standard RL approaches that rasterize spatial structures into grids for CNNs, this method operates directly on triangulated obstacle geometry. It partitions the space using an ego-centric hexagonal grid and utilizes a multi-head attention-based set-encoder to aggregate primitives into a bounded, fixed-size representation. By keeping the RL backbone fixed and matching conditions, the authors demonstrate that this geometry-first interface is highly competitive and outperforms rasterized map representations, particularly in unbounded procedural training regimes where memorizing limited environments is impossible.

**Key Strengths:**
- Sound and tightly controlled experimental design
- Systematic procedural generation ablations (bounded vs. unbounded continuous training)

**Key Weaknesses:**
- The functional components (PMA, hexagonal lattices, MHA, cross attention) are technically standard and combined in a fairly expected manner for spatial data
- The fully observable setup (having access to global maps without requiring exploration) pulls the domain strictly into the sphere of model-based spatial pathfinding, which deviates from standard view-based real-world robot exploration
- Several notational gaps and ambiguities in the mathematical definitions

**Audience:**

Yes

**Audience Explanation:**

The TMLR community, especially those working on representation learning and path planning, would be interested in seeing positive experiments that operating directly on set-valued geometric primitives can bypass the representation bottlenecks and resolution trade-offs caused by rasterized CNNs at scale.

**Claims And Evidence:**

Yes

**Claims Explanation:**

The evidence provided is convincing due to well-isolated ablation configurations. By maintaining a constant RL algorithm across tests and incorporating enriched raster controls (adding SDF and dynamic global/local ego streams), the authors successfully pinpoint the performance deltas. The results correctly exhibit that retaining explicit geometric topologies matters significantly more when generalizing to continuously generated structures.

**Requested Changes:**

- **Missing definitions for navigation vector components:** $\theta_t$ is undefined in the main text. Also, please explicitly state what distance the variable $\hat{L}_t$ measures in Section 3.2.1, rather than just calling it a "distance"
- **Rotation matrices & Baseline comparison:** In $a_t = R_t a_t^{ego}$, $R_t$ is undefined. Presumably, it is the rotation matrix from the goal-aligned ego frame to the world frame. Further, since you rotate observations to align the $x$-axis to the goal, how does this explicit orientation compare to how rasterization baselines perceive heading? It would be good to state whether the CNN baselines implicitly get the exact same relative orientational alignment
- **Equation 5 notation:** Equation 5 currently reads $PMA_k(Z) = MHA(Q=S, K=Z, V=Z) \in \mathbb{R}^{k \times d}$. This index is a bit clunky. It might be clearer to define it as $PMA_k(Z) = MHA(Q=S_k, K=Z, V=Z) \in \mathbb{R}^{d}$ for a given k since $S \in \mathbb{R}^{k \times d}$
- **Variable-length operations vs Fixed tensors:** It's nice to have all those variable-length operators, but you essentially note that you work on a fixed-sized padded tensor ensemble anyway ("M active cells and at most K_cell triangles"). Readers might wonder: is PMA actually necessary at all then? It would strengthen the writing to briefly clarify that despite the zero-padded fixed bounding, the *true* number of elements within is still variable, making permutation-invariant operators + masking critical to avoid overfitting
- **Link to Fully-Observable Planning:** If the methods can process the global map without having to actively explore the physical space to see boundaries, this setup is functionally closer to classical algorithmic planning rather than embodied visual exploration. I strongly suggest discussing this context and explicitly connecting your method to Value Iteration Networks (VIN) and similar fully-observable model-based planning mechanisms in reinforcement learning

Optional:

- **Clarify primitive encoding:** For Figure 4 and the relevant text, briefly clarify why $V_t$ doesn't include the mask $\hat{m}^{tri}_t$.

---

> ### Author Response · Authors · 2026-06-14
>
> We appreciate the reviewer's careful and precise read; the comments sharpened the presentation throughout. All paper changes are marked in blue.
>
> **R1. Navigation vector definitions.** Section 3.2.1 now defines every component of z_t: the heading encoding (cos theta_t, sin theta_t), the agent position (x_t, y_t) in world coordinates, the ego-frame goal coordinates (with the footnoted remark that g~_y = 0 in the goal-aligned frame), and the distance, now stated explicitly as the Euclidean agent–goal distance normalized by the lattice circumradius s.
>
> **R2. Rotation matrix and baseline orientation parity.** Sec. 3.2.1 now defines theta_t and names R_theta as the ego-to-world rotation; App. A.1 gives the explicit matrix, theta_t = atan2(g_y − y_t, g_x − x_t), and notes that the frame is time-dependent (recomputed every step), with a new annotated figure (App. A.1) now referenced from the main text (Sec. 3.2.1). On baseline parity: the raster local crop is rotated by the identical agent–goal heading so that the goal direction aligns with the positive x-axis, and all methods receive the same navigation-vector format. This is stated in the main text (Sec. 4.3) and App. B.2, and a new figure (App. B.2) depicts both streams and their channels, including the goal-aligned rotation of the local crop. Orientation handling is therefore matched across interfaces and does not confound the comparison.
>
> **R3. Equation 5 (PMA) notation.** We genuinely considered adopting the reviewer's per-seed notation; it is clean and elegant, and we went back and forth on it more than once. In the end we kept our original operator-level (set-to-set) form, because it mirrors how PMA actually operates and is implemented: all seeds attend to the cell jointly and their k output tokens together form one multi-faceted summary (k the number of seeds, S the seed matrix, output in R^{k×d}), rather than each seed being read out in isolation. We did take the reviewer's underlying point that the original was ambiguous and addressed it directly: the revision specifies the masked variant explicitly and adds a sentence (Sec. 3.2.2) on why permutation-invariant pooling over the padded set matters (see R4).
>
> **R4. Is PMA necessary given fixed padded tensors?** We make this explicit in two places. In Sec. 3.2.2 we state that, although the tensors are zero-padded to fixed bounds, the true number of valid entries varies from state to state, and masked permutation-invariant pooling keeps the encoder independent of slot order and padding. The Discussion develops the same point and connects it to the deployment-time truncation experiment (App. D.3), already part of the original submission and now referenced directly from the main text. Because the encoder is a masked set operator, the per-cell capacity can be changed at inference time without retraining, and performance degrades only gradually. Per-cell primitive counts vary widely in practice (p50 ranging from 16 to 201 across generators; see the generator-statistics table), and the training families are chosen so that truncation actually occurs in the data (Sec. 4.1), so this invariance is exercised, not hypothetical.
>
> **R5. Connection to VIN / fully observable planning.** We added this connection in the Discussion: we compare the setting to differentiable planning architectures such as Value Iteration Networks (Tamar et al., 2016). VIN-style methods build explicit planning computation over a discretized, fully observed grid MDP, whereas we keep the planner generic and study how a continuous geometric map should be exposed to an RL policy. We also clarify that, despite full map availability, the policy acts through a bounded, lossy observation (active-cell selection and capacity truncation; Sec. 3.2.1), so the setting sits between full-state planning and view-based perception.
>
> **Optional — why the primitive encoding carries no mask.** The validity mask is carried separately by design: the per-triangle encoder is applied independently to each slot, so padded slots never influence other slots; the mask first takes effect in the masked PMA attention, where padded entries are excluded from keys and values and thus cannot contribute to any readout token.

---

### Author Response · Authors · 2026-06-14

We are grateful to all three reviewers for the careful, constructive reviews; they genuinely improved the paper. All changes in the updated PDF are marked in blue.

**1. Truncation robustness (GMXL, crBS).** A deployment-time truncation study, already part of the original submission, is now anchored from the main text (App. D.3): reducing the per-cell primitive capacity at inference time, without retraining, removes triangles from the observation, and performance degrades marginally rather than catastrophically. A new filmstrip figure (App. D.3) shows the partial, capacity-limited view the policy actually receives along successful trajectories.

**2. Partial observability and the link to VIN (7CvC, GMXL, crBS).** Although the underlying map state is lossless, the policy never sees it in full: active-cell selection and per-cell truncation leave it a bounded, lossy view of the scene (Sec. 3.2.1, Discussion; visualized in the Teotihuacán filmstrip, App. D.3). The agent therefore acts under effective partial observability, and still outperforms the raster CNN. In the Discussion we relate this to Value Iteration Networks (Tamar et al., 2016): VIN-style methods assume a discretized, fully observed grid MDP, whereas ours is continuous and only partially observed. We stress, however, that the aim of this paper is not to propose a planner but to study the observation interface itself, i.e., what geometric representation an RL policy should act on; the VIN comparison is therefore about representation, not planning.

**3. Observation interface, clarified and depicted (7CvC, crBS).** We expanded the description of both observation interfaces and added figures making their information content explicit: an annotated hex observation-construction figure showing the goal-aligned ego frame, the lattice, and per-cell triangle binning (App. A.1), and a raster-streams figure depicting both raster streams (local crop and global map) and their channels — occupancy, SDF, goal Gaussian (App. B.2). With the component-by-component definition of the navigation vector (Sec. 3.2.1), these make precise what each method receives as input.

**4. Definitions and notation (7CvC, crBS).** Every component of the navigation vector z_t is now defined explicitly, including the agent-position convention and the normalized agent–goal Euclidean distance (defined in Sec. 3.2.1). The heading theta_t and the ego-to-world rotation R_theta are defined in the main text, with the explicit matrix and the time-dependence of the ego frame in App. A.1. The PMA definition is clarified in context: the masked variant is specified, and a new sentence explains why permutation-invariant pooling over the padded tensors is essential.

**5. Experimental-protocol completeness (crBS).** The reset-difficulty parameterization is fully specified (closed-form schedules, curriculum sampling distribution, and controller; App. C.1, Fig. C.1) and referenced from the main text. "Asymptotic" is defined precisely (mean over the final 10% of training); "curriculum mean" is the controller mean mu_k. The bounded-regime pool size is stated (101 generation seeds per training family). Baseline details were expanded: no recurrence (LSTM) or distributed training is used for any method; IMPALA/IMPOOLA refer to the convolutional encoders, which are standard in deep RL (Espeholt et al. 2018; Trumpp et al. 2025); the Transformer baseline and Hex-Attn are now explicitly characterized.

**6. Claim calibration (crBS).** Path-efficiency claims in the bounded regime were softened: differences fall within seed variability, and we no longer claim the strongest overall path efficiency there.

**7. Hexagonal parcellation (crBS).** A new Discussion paragraph explains the choice (including the grid-cell connection the reviewer suggested, Moser et al. 2014) while explicitly not claiming the hexagonal shape is load-bearing. It is a sensible choice, as a hexagonal lattice covers the receptive field with fewer cells than a square grid.